

# Bootstrapping traceless symmetric $O(N)$ scalars

**Marten Reehorst[1,2], Maria Refinetti[1,3] and Alessandro Vichi[1,4]**

**1** Institute of Physics, École Polytechnique Fédérale de Lausanne (EPFL),
Rte de la Sorge, BSP 728, CH-1015 Lausanne, Switzerland
**2** Institut des Hautes Études Scientifiques, Bures-sur-Yvette, France
**3** Laboratoire de Physique de l'École Normale Supérieure, Université PSL, CNRS,
Sorbonne Université, F-75005 Paris, France
**4** Dipartimento di Fisica dell'Università di Pisa and INFN,
Largo Pontecorvo 3, I-56127 Pisa, Italy

## Abstract

We use numerical bootstrap techniques to study correlation functions of traceless symmetric tensors of $O(N)$ with two indices $t_{ij}$. We obtain upper bounds on operator dimensions for all the relevant representations and several values of $N$. We discover several families of kinks, which do not correspond to any known model and we discuss possible candidates. We then specialize to the case $N = 4$, which has been conjectured to describe a phase transition in the antiferromagnetic real projective model $ARP^3$. Lattice simulations provide strong evidence for the existence of a second order phase transition, while an effective field theory approach does not predict any fixed point. We identify a set of assumptions that constrain operator dimensions to a closed region overlapping with the lattice prediction. The region is still present after pushing the numerics in the single correlator case or when considering a mixed system involving $t$ and the lowest dimension scalar singlet.



# 1 Introduction

The conformal bootstrap [1, 2] (see [3, 4] for a review) has successfully classified many 3D CFTs, providing stringent predictions of operator dimensions, which translate in precise determinations of the corresponding critical exponents [5–11]. These techniques have been used to study many problems including multiple scalars [12–17], fermions [18–20], currents [21, 22], stress tensors [23] and various global symmetry representations [24–44].

In this work we push this program further and explore the space of three dimensional conformal field theories (CFTs) containing a scalar operator $t_{ij}$, which is a traceless symmetry tensor of $O(N)$ with rank-2. While such operators are also present in the well studied $O(N)$-vector models, here we want to target fixed points of gauge theories, where the operator $t_{ij}$ can arise as the simplest gauge invariant scalar made from more elementary fields, charged under the gauge symmetry.

Similar studies have been done for adjoint representations of $SU(N_f)$ in four dimensions, with application to the conformal window of QCD-like theories. In that case, however, bootstrap bounds have not revealed any surprise [37, 38]. On the contrary, the present setup will show many interesting features.

In addition to the general exploration of CFTs, in the present work we also address the existence of a fixed point observed in the antiferromagnetic real projective model with $N$ components $ARP^{N-1}$, in the specific case $N = 4$. Lattice simulations present strong evidences of a second order phase transition, driven by an order parameter transforming in the rank-2 representation of $O(4)$; on the contrary, an effective approach based only on the Landau-Ginzburg-Wilson paradigm seems to disagree [45]. We will present bootstrap evidences confirming the existence of a fixed point. We will also discuss new prediction for certain operator dimensions and OPE coefficients that could be tested by future lattice studies.

Before entering in the bootstrap setup and present our results, let us broadly discuss what theories must be consistent with our bootstrap bounds. The following analysis will also guide us through the choice of reasonable assumptions to isolate theories of interest.

## 1.1  $RP^{N-1}$ and $ARP^{N-1}$ models

We begin with a simple lattice model, the $(A)RP^{N-1}$, which is defined as a system of spins $\mathbf{s_x}$ taking values in the real projective space $RP^{N-1}$, with the index $\mathbf{x}$ labelling the lattice site. Equivalently, we can describe the system by considering $\mathbf{s_x}$ to take values in $\mathbb{R}^N$, with the restriction $\mathbf{s_x} \cdot \mathbf{s_x} = 1$ and the identification $\mathbf{s_x} \sim -\mathbf{s_x}$; the latter condition can be viewed as a $\mathbb{Z}_2$ gauge symmetry, since one can change sign to each spin independently, i.e. locally. The hamiltonian can be written as

$$H_{RP^{N-1}} = J \sum_{\langle \mathbf{x}, \mathbf{y} \rangle} \left| \mathbf{s_x} \cdot \mathbf{s_y} \right|^2 , \tag{1}$$

where $\langle \mathbf{x}, \mathbf{y} \rangle$ indicates that the sum runs over pairs of nearest neighbors. For negative $J$ the system is ferromagnetic while for positive $J$ it is antiferromagnetic. This model has been studied in the antiferromagnetic regime and for $N \leq 4$ using lattice simulations [45]. It was found that for $N = 2, 3$ the IR admits a second order phase transition, and the IR fixed point seems to be in the same universality class of the $O(2)$ and $O(5)$ model respectively. The case $N = 4$ is particularly interesting, since it still presents evidences of a second order phase transition but this time the critical exponents do not correspond to those of the $O(m)$-model, for any $m$. Moreover the transition appears to be driven by an order parameter transforming in the traceless symmetric representation of $O(4)$.

Let us briefly discuss the structure of the order parameter, as it will be useful also for the discussion in the next sections. In the *ferromagnetic* case, the energy is minimized by aligning the directions of the spins. Thus, at low energy the system breaks $O(N)$ symmetry by aligning in a preferred direction. This configuration preserves translational invariance. In the standard LGW approach one looks for a gauge invariant order parameter that is non-zero in the ordered phase and vanishes in the disordered phase. This order variable is built from the *site variable*, $P_x^{ab} = s_x^a s_x^b - \delta^{ab}/N$. We then define the order parameter as its sum over lattice sites $M^{ab} = \sum_x P_x^{ab}$. We see that in the ordered phase the contributions to $M^{ab}$ are cumulative, due to the preferred direction, resulting in a non-zero matrix. At high temperature, in the isotropic phase, contributions will cancel so that $M^{ab} \to 0$ in the infinite temperature limit. This order parameter transforms as a traceless symmetric representation of $O(N)$ and is invariant under a lattice symmetry that interchanges two sublattices.[1]

In the *antiferromagnetic* case the energy is instead minimized by taking $\mathbf{s_x} \cdot \mathbf{s_y} = 0$ for neighboring sites. Thus, in the ordered phase every spin is orthogonal to its nearest neighbor. Unlike anti-correlation in the usual ferromagnetic case, here one can divide the lattice in two sublattices, and the spins are orthogonal among the two. Orthogonality does not fix the configuration uniquely unlike correlation or anti-correlation. Thus, it is not immediately clear what the symmetries of the ordered state are and what order parameter has a non-zero expectation value in the ordered phase. In [46], for the similar case of $CP^2$, it was shown that the order parameter must also break the symmetry that interchanges the sublattices. This proof can easily be extended to the case of $ARP^2$. Unfortunately we don't know of any proof for $N > 2$. If we assume the same holds for general $N$ the correct order parameter is built from a staggered site variable $A_x^{ab} = p_x P_x^{ab}$, where $p_x = \exp\left[i\pi \sum_{k=1}^3 x_k\right]$, i.e. the parity of the lattice site. Summing over the staggered site variable the order parameter is given by $M^{ab} = \sum_{\mathbf{x}} A_{\mathbf{x}}^{ab}$. This order parameter also transforms as a traceless symmetric representation of $O(N)$ but this time is odd under the $\mathbb{Z}_2$ symmetry.

---

[1]Gauge invariance forbids a linear order parameter $s_x^a$ so the next simplest order parameter is quadratic. The vanishing of the order parameter in the disordered phase forces the subtraction of the trace resulting in the traceless symmetric representation.

The lattice analysis[2] for $ARP^3$ led to the following estimates of the critical exponents:

$$\Delta_s = 3 - \frac{1}{\nu} = 1.28 \pm 0.13\,, \quad \Delta_t = \frac{1+\eta}{2} = 0.54 \pm 2\,, \quad \Delta_{s'} > 3 \quad \text{(lattice results [45])}\,. \quad (2)$$

## 1.2   The Landau-Ginzburg-Wilson effective action

In many cases of physical interest one can understand the critical behavior of a lattice system also starting from a UV description in terms of a field theory of a scalar field with only a few renormalizable interactions. Thanks to the properties of the RG flow, if the two UV theories belong to the same universality class, they will flow to the same fixed point in the IR.

Physically this is equivalent to identifying the order parameter that describes the fluctuations near criticality and writing an effective Hamiltonian. The order parameter is chosen such that it vanishes in the disordered phase and is non-zero in the ordered phase. Thus, it is expected to be small near criticality and it make sense to consider only the leading terms.

If one is interested in describing the phase transition observed for $ARP^{N-1}$, the order parameter $\Phi_{ij}$ is a traceless symmetric rank-2 tensor of $O(N)$, odd under an additional $\mathbb{Z}_2$ symmetry. The LGW Hamiltonian reads:

$$\mathcal{H} = \text{Tr}\big(\partial_\mu \Phi\big)^2 + r\,\text{Tr}\,\Phi^2 + u_0(\text{Tr}\big(\Phi^2\big))^2 + \frac{v_0}{4}\,\text{Tr}\,\Phi^4\,. \quad (3)$$

The analysis of the $\beta$-functions for the couplings $u_0$ and $v_0$ in $\varepsilon$-expansion at one loop reveals the existence of four fixed points. Two of them are well known: the free Gaussian theory ($u_0^* = v_0^* = 0$) and the $O(N')$ Wilson-Fisher fixed point ($v_0^* = 0$), with $N' = N(N+1)/2 - 1$ the total number of scalars encoded in the tensor $\Phi$. In addition, there are two fixed points, with both coupling non-zero, that merge at $N = N_c$ and turn complex for $N > N_c$. A Borel resummation of the five-loop $\varepsilon$-expansion predicts $N_c \approx 3.6$ [45]. For $N = 2, 3$ the additional relation $\text{Tr}\,\Phi^4 = (\text{Tr}\,\Phi^2)^2/2$ holds. So even for $N < N_c$ the new fixed points can be mapped respectively to the $O(2)$ and $O(5)$ model. In conclusion, the LGW analysis predicts that no fixed point exist for this model besides the WF ones. This is in tension with the lattice results discussed in the previous section.

## 1.3   Scalar gauge theories

Traceless symmetric tensors of $O(N)$ can arise in a many different theories. A general bootstrap analysis will be sensitive to all of them. As an example, in this section we review the known results for a model based on a theory with local $O(M)$ gauge invariance and a global $O(N)$ symmetry (see for instance [47, 48] and the references therein). The lagrangian for such a model is given by:

$$\mathcal{L} = -\frac{1}{4}F^a_{\mu\nu}F^{a\mu\nu} + \frac{1}{2}\sum_{i=1}\big(D_\mu\phi_i\big)^\alpha(D^\mu\phi_i)^\alpha + V(\phi_i^\alpha)\,,$$

$$\big(D_\mu\phi_i\big)^\alpha = \partial_\mu\phi_i^\alpha - (T^b)^\alpha{}_\beta\phi_i^\beta A_\mu^b\,, \qquad V(\phi_i^\alpha) = u_0 S^2 + v_0\sum_{i,j}Q_{ij}Q_{ij}\,,$$

$$S = \sum_{a,k}(\phi_k^\alpha\phi_k^\alpha)\,, \qquad Q_{ij} = \sum_a \phi_i^\alpha\phi_j^\alpha - \frac{1}{N}\delta_{ij}S\,. \quad (4)$$

---

[2]The analysis of [45] used finite-size-rescaling to study the RG invariant $R_\xi = \frac{\xi}{L}$, where $\xi$ is the correlation length and $L$ the lattice's size. It is observed that lines of different $L$'s meet at a critical temperature $\beta_c = 6.779(2)$ and the critical exponent $\nu = 0.59(5)$ was estimated. The error is due to different methods of fitting the data, while the statistical error is much smaller. Moreover, they were able to extract the critical exponent $\eta = 0.08(4)$ by analyzing the behavior of the susceptibility around the fixed point. Finally, a study of the Binder parameter shows sizeable corrections due to scaling possibly indicating an un-tuned singlet with a dimension that is close to relevant. However, the data was insufficient to give a reliable estimate on the corresponding critical exponent.

where $\mu$ and $\nu$ are spacetime indices, $\alpha, \beta, \gamma = 1, \ldots, M$ are fundamental indices of the gauge group $O(M)$, $a, b, c = 1, \ldots, M(M-1)/2$ are adjoint indices and $i, j, k = 1, \ldots, N$ are indices of the global flavor group. The presence of a gauge symmetry imposes that, at the fixed points, local operators must be made from gauge invariant combinations of the fields $\phi_i^\alpha$ and the field strength $F_{\mu\nu}^a$. In particular the smallest dimensions scalars are the singlet $S$ and the traceless symmetric $O(N)$ tensor $Q_{ij}$ defined in (4).

The above models have been extensively studied: the $\varepsilon$-expansion [48] predicts the existence of a fixed point only for

$$N > 44(M-2). \tag{5}$$

Moreover, the $\varepsilon$-expansion shows that the gauge invariant model is always stable compared to the enhanced $O(NM)$ model. Alternatively, one can study the model in 3d, in the large-$N$ limit at fixed $M$. For instance one obtains [47]:

$$\Delta_S = 1 + \frac{16}{3\pi^2 N}(9M-7) + O\left(\frac{1}{N^2}\right),$$
$$\Delta_Q = 1 - \frac{16}{3\pi^2 N}(3M-5) + O\left(\frac{1}{N^2}\right). \tag{6}$$

Clearly the above expressions cannot be trusted at small values of $N$. Nevertheless one could compare these expressions with the bootstrap bounds. The main issue is that, given $N$, there are in principle infinitely many underlining gauge theories with the same global symmetry but different CFT-data, as shown already by the leading corrections in Eq. (6).[3]

Let us conclude this overview by discussing a few basic differences among the theories discussed so far. First of all, in presence of a continuous gauge symmetry, the spectrum of the CFT will be richer, given the presence of extra states such as glue-balls $(F_{\mu\nu}^a)^2$ or combination of the two fundamental fields.[4] On the contrary, if the gauge symmetry is discrete, as is the case for the discrete $\mathbb{Z}_2$ gauge symmetry of $RP^N$ models, we do not expect these extra states.

Interestingly, this is not the only difference. Consider for instance the smallest operator transforming in the representation described by a squared Yang-tableau with four boxes, ⊞. We call it the *Box* representation. We will see in the next section that such representation appears in the OPE of two rank-2 tensors. In a gauge theory like in (4), the smallest scalar in the Box representation is given by

$$\mathcal{O}_{ij,kl} \sim Q_{ik}Q_{jl} - Q_{il}Q_{jk} - \text{traces}. \tag{7}$$

The non-triviality of this operator is guaranteed by the internal gauge indices. However, if these were absent, one could not construct it: given a real scalar operator $s_i$ the smallest non trivial operator in the Box representation that one can construct requires two derivatives

$$\mathcal{O}'_{ij,kl} \sim J_{ik}^\mu J_{\mu jl}, \qquad J_{ij}^\mu = s_i \partial^\mu s_j - s_j \partial^\mu s_i, \tag{8}$$

or more fields. This reasoning is valid only in a neighborhood of the UV description, however it gives us an intuition about which operators we should expect in the CFT. Hence, we do not expect the IR fixed point of $(A)RP^N$ models to have light scalars in the Box representation.

More in general, the impossibility to construct light operators in a given representation can be a guiding principle to distinguish different theories, especially when gauge symmetries are involved. Let us view another example: in the LGW model the fundamental field

---

[3]Note that (6) has been obtained in the limit of large $N$, while keeping $M$ fixed. If instead one consider $M \sim N$ then the expansion would change.

[4]Only a subset of those operators, such as glueballs, are accessible with the bootstrap setup considered in this paper.

is a traceless symmetric tensor, while in a gauge theory the fundamental field is a vector of $O(N)$, with an additional gauge index. Although $\phi_i^\alpha$ is not gauge invariant, the existence of a more fundamental building block has important consequences and does have an impact on the spectrum of the CFT. For instance, it is possible to construct a barion-like state of the form $B_{[i_1 \cdots i_M]} \sim \epsilon_{\alpha_1 \cdots \alpha_M} \phi_{[i_1}^{\alpha_1} \cdots \phi_{i_M]}^{\alpha_M}$, transforming in the antisymmetric representation with $M$ indices of $SO(N)$.[5] This has a small dimension for small values of $M$. In the LGW theory the lightest state in same representation would be much heavier.

Finally, a major difference between the gauge model (4) and the LGW description is that the latter displays a $\mathbb{Z}_2$ symmetry in the UV, while the former doesn't. From the CFT point of view, this symmetry imposes the vanishing of three point functions $\langle \Phi_{ij} \Phi_{kl} \Phi_{rs} \rangle$ in a putative fixed point of the LGW model, while the correlator $\langle Q_{ij} Q_{kl} Q_{rs} \rangle$ is allowed to be non-zero in the model based on a gauge theory.

## 2 Setup

In this section we explain the bootstrap setup of the $\langle tttt \rangle$ correlator and its extension to the mixed $t-s$ bootstrap. We first discuss the operators that can be exchanged in the $t \times t$ OPE. We then explain how to write the crossing equations and the resulting sum rules for the single $\langle tttt \rangle$ correlator. Next we present the extension to the mixed $t-s$ bootstrap. In appendix C we also show how this bootstrap setup for the traceless symmetric bootstrap of $O(N)$ is related to the vector bootstrap of $O(N')$ with $N' = N(N+1)/2 - 1$.

### 2.1 The $t \times t$ OPE

We can write the $t \times t$ OPE as

$$t_{\square\square} \times t_{\square\square} = \sum_{\Delta, l} \lambda_{\Delta,l}^S S + \lambda_{\Delta,l}^{T^2} T_{\square\square}^2 + \lambda_{\Delta,l}^{T^4} T_{\square\square\square\square}^4 + \lambda_{\Delta,l}^{A^2} A_{\square}^2 + \lambda_{\Delta,l}^H H_{\square\square\square} + + \lambda_{\Delta,l}^B B_{\square\square}. \quad (9)$$

Here $S$, $T^2$, $T^4$, $A^2$ refer respectively to the singlet, traceless symmetric, four-index symmetric and the antisymmetric representations. $H$ refers to the mixed symmetry $\{3,1\}$ representation which we will call Hook representation, while $B$ refers to the $\{2,2\}$ representation or Box representation. In the rest of the paper we will leave out the young tableau notation and refer to a dimension $\Delta$ and spin $l$ operator as $R_{\Delta,l}$, where $R \in \{S, T^2, T^4, A^2, H, B\}$.

Important special cases of operators are the first antisymmetric vector, i.e. the conserved current $J = A_{2,1}^2$, the first spin-two singlet, i.e. the stress tensor $T = S_{3,2}$. The first antisymmetric vector after the current will be denoted $J'$ and the first spin-2 singlet after the stress tensor $T'$. Furthermore, we will refer to the first singlet scalar as $s$ and the external traceless symmetric scalar as $t$. Again higher dimensional operators will be referred to by adding primes. For example $s'$ refers to the second lowest dimensional singlet operator. $t'$ will denote the first traceless symmetric operator other than $t$-itself. Similarly, the first scalar in the Box representation and the first vector in the Hook representation will be denoted by $b$ and $h$ respectively.

Under exchange of $x_1$ and $x_2$ the spatial part of the three point function $\langle t(x_1)t(x_2)O_{\Delta,\ell}(x_3) \rangle$ goes to $(-1)^\ell$ times itself. Thus, for even spins the global tensor structure must be symmetric under the exchange of the indices of the first and second operator, and antisymmetric for odd spins. The $\{S, T^2, T^4, B\}$ representations only allow a symmetric structure while the $A$ and $H$ representations only allow an antisymmetric tensor structure. Thus,

---

[5]This operator is invariant under $SO(M)$ and not the full $O(M)$ symmetry, however our bootstrap setup does not distinguish between $SO(M)$ and $O(M)$ symmetries so such a theory could also show up in our bounds.

the former set of representations will be exchanged for even spin and the latter set for odd spin.

Two OPE coefficients are of special interest. Ward identities relate the OPE coefficients of stress tensor $T$ and the conserved current $J$ respectively to the central charges $C_J$ and $C_T$:

$$\frac{C_{J_{\text{free}}}}{C_J} = \lambda_{ttJ}^2 \,, \tag{10}$$

$$\frac{C_{T_{\text{free}}}}{C_T} = \frac{\lambda_{ttT}^2}{\Delta_t^2} = \frac{\lambda_{ssT}^2}{\Delta_s^2} \,. \tag{11}$$

In order to construct the correct $O(N)$ tensor structures for 3 and 4pt functions we used an index free notation similar to the one introduced for spacetime indices in [49]. The young tableaux describing the $O(N)$ irreps illustrate how indices corresponding to blocks appearing in the same row are symmetrized while blocks appearing in the same column are anti-symmetrized. The symmetrization of any row can automatically be enforced by contracting all indices corresponding to the same row with the same polarization vector $S$. Similarly, indices corresponding to the next row are contracted with $U$ and so on (in this paper no irreps with more than two rows appear). One then only needs to enforce the anti-symmetry and tracelessness by hand. We review in details our method in appendixes A and B.

## 2.2 4pt functions and the crossing equations

The crossing equations are obtained in the standard way by equating the s-channel and t-channel decompositions of the 4pt-function. The 4pt-function $\langle tttt \rangle$ has six independent tensor structures, each providing a crossing equation of the form

$$\sum_{R,\mathcal{O}_R} \lambda_{12\mathcal{O}_R} \lambda_{34\mathcal{O}_R} \frac{g_{\Delta_{\mathcal{O}_R},\ell_{\mathcal{O}_R}}^{\Delta_{12},\Delta_{34}}(z,\bar{z})}{(z\bar{z})^{\frac{\Delta_1+\Delta_2}{2}}} = \sum_{R',\mathcal{O}_{R'}'} \lambda_{32\mathcal{O}'} \lambda_{14\mathcal{O}'} \frac{g_{\Delta_{\mathcal{O}_{R'}'},\ell_{\mathcal{O}_{R'}'}}^{\Delta_{32},\Delta_{14}}(1-z,1-\bar{z})}{((1-z)(1-\bar{z}))^{\frac{\Delta_3+\Delta_2}{2}}} \,. \tag{12}$$

Here $z$ and $\bar{z}$ are the standard crossing ratios and $g$ is the scalar conformal block. For the single correlator (of identical operators) both $R$ and $R'$ run over $\{S, T^2, T^4, A^2, H, B\}$ and $\Delta_{ij} = 0 \,\forall\, i, j$.

The final crossing equations for $\langle tttt \rangle$ can be written as

$$\sum_{\mathcal{O}} \lambda_{\mathcal{O}}^2 V_{S,\Delta,\ell} + \sum_{\mathcal{O}} \lambda_{\mathcal{O}}^2 V_{T^2,\Delta,\ell} + \sum_{\mathcal{O}} \lambda_{\mathcal{O}}^2 V_{T^4,\Delta,\ell} + \sum_{\mathcal{O}} \lambda_{\mathcal{O}}^2 V_{B,\Delta,\ell} + \sum_{\mathcal{O}} \lambda_{\mathcal{O}}^2 V_{A,\Delta,\ell} + \sum_{\mathcal{O}} \lambda_{\mathcal{O}}^2 V_{H,\Delta,\ell} = 0_{1\times 6} \,, \tag{13}$$

where $V_{R,\Delta,\ell}$ is a 6 dimensional vector describing the contribution of a primary operator $\mathcal{O}$ of dimension $\Delta$, spin $\ell$, and representation $R$. The vector $V_{R,\Delta,\ell}$ is expressed in terms of the usual $F$'s and $H$'s

$$\begin{aligned}
H &= u^{\frac{1}{2}(\Delta_2+\Delta_3)} g_{\Delta,\ell}^{\Delta_{12},\Delta_{34}}(v,u) + v^{\frac{1}{2}(\Delta_2+\Delta_3)} g_{\Delta,\ell}^{\Delta_{12},\Delta_{34}}(u,v) \,, \\
F &= v^{\frac{1}{2}(\Delta_2+\Delta_3)} g_{\Delta,\ell}^{\Delta_{12},\Delta_{34}}(u,v) - u^{\frac{1}{2}(\Delta_2+\Delta_3)} g_{\Delta,\ell}^{\Delta_{12},\Delta_{34}}(v,u) \,.
\end{aligned} \tag{14}$$

Here $g^{\Delta_{12},\Delta_{34}}$ is the scalar conformal block normalized as entry 1 of Table I in [3]. In this section the only correlation under consideration is $\langle tttt \rangle$ and this simplifies to

$$\begin{aligned}
H &= u^{\Delta_t} g_{\Delta,\ell}(v,u) + v^{\Delta_t} g_{\Delta,\ell}(u,v) \,, \\
F &= v^{\Delta_t} g_{\Delta,\ell}(u,v) - u^{\Delta_t} g_{\Delta,\ell}(v,u) \,.
\end{aligned} \tag{15}$$

The crossing equations can also be represented by a 6 by 6 matrix. Its explicit form is[6]

$$M_{\langle tttt \rangle, O(N)} = \begin{pmatrix} F & 0 & 0 & 0 & \frac{1}{2}F(N+4)(N-1) & -FN \\ 0 & F & 0 & 0 & \frac{1}{2}F(N-2) & -\frac{FN}{2} \\ 0 & 0 & -F & 0 & \frac{1}{2}F(N+4) & -\frac{1}{2}F(N+2) \\ 0 & 0 & 0 & F & -3F & 2F \\ H & 0 & -\frac{2H(N-1)}{N} & -\frac{H(N+4)(N+6)(N-1)}{12N} & -\frac{H(N+4)(N-2)(N-1)}{4N} & -\frac{H(N+2)(N-3)(N-2)}{6N} \\ 0 & H & -\frac{H(N+4)(N-2)}{N(N+2)} & -\frac{H(N+6)(N-2)}{3N} & \frac{H(N+4)(N-2)}{N(N+2)} & \frac{H(N+4)(N-3)}{3N} \end{pmatrix}. \quad (16)$$

Here rows correspond to the six different equations and columns correspond to the vectors $\{V_S, V_{T^2}, V_A, V_{T^4}, V_H, V_B\}$ in equation 13. The bootstrap problem consists of finding a positive linear functional $\alpha$ such that

$$\begin{cases} \alpha(V_{\mathbb{I}}) = 1, \\ \alpha(V_R) \geq 0 \qquad \forall R \in \{S, T^2, T^4, A^2, H, B\}, \quad \forall \Delta_{R,\Delta,\ell} > \Delta^*_{R,\Delta,\ell}. \end{cases} \quad (17)$$

If such a functional exists it excludes a spectrum with $\Delta_{R,\Delta,\ell} > \Delta^*_{R,\ell}$. $\Delta^*_{R,\Delta,\ell}$ is usually taken to be the unitarity bound except when we try to find the maximal allowed gap for a certain operator or when we have reason to assume a gap above the unitarity bound for a theory that we are trying to isolate.

In practice the crossing equations are truncated by taking derivatives around the crossing symmetric point $z = \bar{z} = 1/2$ and the maximal number of derivatives is denoted by $\Lambda$. These truncated crossing equations are used as input in the arbitrary precision semi-definite programming solver SDPB (version 2) [50, 51]. The truncations and parameters used in the numerical implementation can be found in tables 2 and 1.The computations were managed using Simpleboot [52].

In addition to finding the feasible set of $\Delta_{R,\Delta,\ell}$ we can also find lower and upper bounds on squared OPE coefficients $\lambda^2_{tt\mathcal{O}}$ by picking the corresponding vector $V_\lambda$ to define the normalization of $\alpha$, i.e. $\alpha(V_\lambda) = \pm 1$ and maximizing the objective $\alpha(V_{\mathbb{I}})$.[7]

## 2.3 Setup of mixed $t-s$ bootstrap

In this section we write the bootstrap equations for the system of correlators involving the traceless symmetric operator $t$ and the leading singlet $s$. We will restrict ourselves to the case in which $t$ is odd under a $\mathbb{Z}_2$ symmetry, since our goal is to study the $ARP^3$ model discussed in section 1.1. In that case the full system of crossing equations is given by the crossing equations of the correlators $\langle ttss \rangle$ and $\langle stts \rangle$, $\langle tsts \rangle$, and $\langle ssss \rangle$. Crossing equations involving three $t$-operators vanish because $t \times s$ can only exchange $\mathbb{Z}_2$ odd operators while $t \times t$ can only exchange $\mathbb{Z}_2$ even operators. All new correlators are constrained to exchange only a single irrep: $s \times s$ can only exchange neutral operators while $t \times s$ can only exchange operators in the $T^2$ irrep. The $t \times s$ OPE does not have the permutation symmetry that the $t \times t$ OPE had and thus allows the exchange of both odd and even spin traceless symmetric operators.

Note that when we do not impose a gap forbidding the exchange of the external operator $t$ in $t \times t$ results using this setup also hold for $\mathbb{Z}_2$-even $t$.[8]

---

[6]The exact form depends on the normalization of the OPE coefficients. We are free to rescale columns by any positive factor and absorb this into the OPE coefficients. We are of course also free to rescale rows, i.e. equations, by any factor.

[7]Normalizing $\alpha(V_\lambda) = 1$ will give us an upper bound on the OPE coefficient, while $\alpha(V_\lambda) = -1$ will give a lower bound.

[8]The inclusion of $\langle ttts \rangle$ would add a new crossing symmetric $O(N)$ tensor structure where only the product of OPE coefficients $\lambda_{tt\mathcal{O}} \lambda_{ts\mathcal{O}}$ enter.

Restricting to the crossing equations for $\mathbb{Z}_2$-odd $t$ there are four additional crossing equations, two between $\langle sstt \rangle$ and $\langle tsst \rangle$, one from $\langle tsts \rangle$ and one from $\langle ssss \rangle$. The crossing equations can now be written as

$$
\sum_{\mathcal{O}} (\lambda_{tt\mathcal{O}} \ \lambda_{ss\mathcal{O}}) V_{S,\Delta,\ell} \begin{pmatrix} \lambda_{tt\mathcal{O}} \\ \lambda_{ss\mathcal{O}} \end{pmatrix} + \sum_{\mathcal{O}_E} \lambda_{tt\mathcal{O}_E}^2 V_{T^2,E,\Delta,\ell} + \sum_{\mathcal{O}_o} \lambda_{ts\mathcal{O}_o}^2 V_{T^2,O,\Delta,\ell} + \sum_{\mathcal{O}} \lambda_{\mathcal{O}}^2 V_{T^4,\Delta,\ell}
$$

$$
+ \sum_{\mathcal{O}} \lambda_{tt\mathcal{O}}^2 V_{B,\Delta,\ell} + \sum_{\mathcal{O}} \lambda_{tt\mathcal{O}}^2 V_{A,\Delta,\ell} + \sum_{\mathcal{O}} \lambda_{tt\mathcal{O}}^2 V_{H,\Delta,\ell} + (\lambda_{tts} \ \lambda_{sss}) V_{\text{ext.}} \begin{pmatrix} \lambda_{tts} \\ \lambda_{sss} \end{pmatrix} = 0_{1 \times 10},
$$

Here we have chosen to separate out the contributions proportional to the OPE coefficients of the external vector into a separate vector $V_{ext.}$. Since the $A$, $T^4$, $H$ and $B$ representations cannot be exchanged in the new correlators the vectors $V_A, V_{T^4}, V_H, V_B$ remain unaffected (apart from padding them with an appropriate number of zeros at the end). The entries of $V_S$ become matrices since there are now contributions proportional to $\lambda_{ttS}^2$, $\lambda_{ttS}\lambda_{ssS}$ and $\lambda_{ssS}^2$. Furthermore, we split the traceless symmetric contribution into a $\mathbb{Z}_2$ even part coming from the $t \times t$ OPE and a $\mathbb{Z}_2$ odd part coming from $t \times s$ OPE. The $\mathbb{Z}_2$ even part remains identical to the vector $V_{T^2}$ in equation 16. The $t \times s$ OPE exchanges traceless symmetric operators of both odd and even spin. The new vectors $V_S$, $V_{T^2,O}$ and $V_{\text{ext.}}$ are given by

$$
V_S = \begin{pmatrix} \frac{1}{2}\big((N+N^2)-2\big)\mathcal{F}_{11}^{\Delta_{tt}\Delta_{tt}} \\ 0 \\ 0 \\ 0 \\ \frac{1}{2}\big((N+N^2)-2\big)\mathcal{H}_{11}^{\Delta_{tt}\Delta_{tt}} \\ 0 \\ 0 \\ -\frac{1}{2}\mathcal{H}_{12}^{\Delta_{ss}\Delta_{ss}} \\ \frac{1}{2}\mathcal{F}_{12}^{\Delta_{ss}\Delta_{ss}} \\ \mathcal{F}_{22}^{\Delta_{ss}\Delta_{ss}} \end{pmatrix}, V_{T^2,O} = \begin{pmatrix} 0 \\ 0 \\ 0 \\ 0 \\ 0 \\ 0 \\ \mathcal{F}^{\Delta_{ts}\Delta_{ts}} \\ (-1)^L \mathcal{H}^{\Delta_{ts}\Delta_{ts}} \\ (-1)^L \mathcal{F}^{\Delta_{ts}\Delta_{ts}} \\ 0 \end{pmatrix}, V_{\text{ext.}} = \begin{pmatrix} \frac{1}{2}\big((n+n^2)-2\big)\mathcal{F}_{11}^{\Delta_{tt}\Delta_{tt}} \\ 0 \\ 0 \\ 0 \\ \frac{1}{2}\big((n+n^2)-2\big)\mathcal{H}_{11}^{\Delta_{tt}\Delta_{tt}} \\ 0 \\ \mathcal{F}_{11}^{\Delta_{ts}\Delta_{ts}} \\ \mathcal{H}_{11}^{\Delta_{ts}\Delta_{ts}} - \frac{1}{2}\mathcal{H}_{12}^{\Delta_{ss}\Delta_{ss}} \\ \mathcal{F}_{11}^{\Delta_{ts}\Delta_{ts}} + \frac{1}{2}\mathcal{F}_{12}^{\Delta_{ss}\Delta_{ss}} \\ \mathcal{F}_{22}^{\Delta_{ss}\Delta_{ss}} \end{pmatrix}, \quad (18)
$$

where we defined the matrices

$$
(\mathcal{F}_{ij}^{\Delta_1,\Delta_2})_{mn} = \begin{cases} F^{\Delta_1,\Delta_2}, & (i=n \wedge j=m) \vee (i=m \wedge j=n), \\ 0, & \text{else,} \end{cases}
$$
$$
(\mathcal{F}_{ij}^{\Delta_1,\Delta_2})_{mn} = \begin{cases} H^{\Delta_1,\Delta_2}, & (i=n \wedge j=m) \vee (i=m \wedge j=n), \\ 0, & \text{else.} \end{cases} \quad (19)
$$

Finally, let us comment that the mixed $t-s$ setup does not break the map between the $O(N')$ vector bootstrap and the $O(N)$ traceless symmetric bootstrap and the same relations between positive functionals described in appendix C still hold.

## 3 A systematic study of general $N$

Here we present a systematic study of bounds on the dimension of the first operator in all representations for general $N$. Specifically we examine $N = 4, 5, 10, 20, 100$ and occasionally $N = 1000$ to study the asymptotic of certain kinks at large $N$. The bounds on the leading operators in the singlet representation are identical to the corresponding bounds found in the $O(N')$-vector bootstrap,[9] where $N' = N(N+1)/2 - 1$. For other representations there is not such relation.

---

[9]This is proven in appendix C.

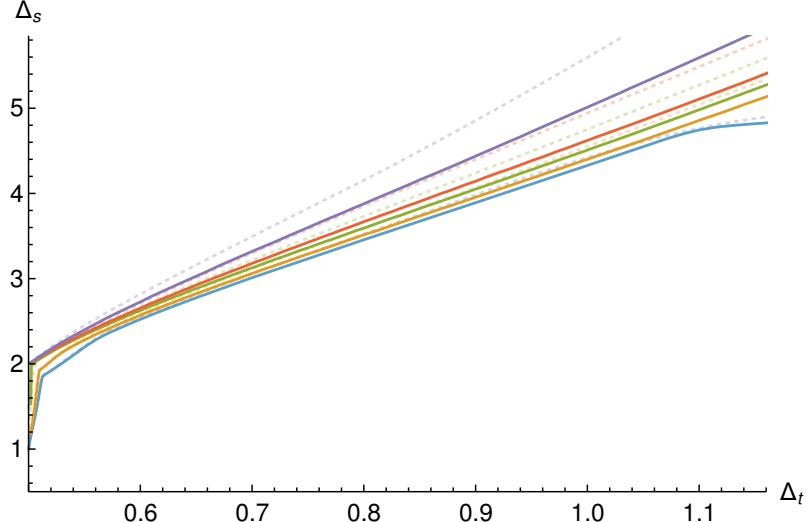

Figure 1: Bound on the dimension of the first singlet scalar. The blue, orange, green, red and purple lines correspond respectively to $N = 4, 5, 10, 20, 100$. These bounds have been obtained at $\Lambda = 27$. The dotted lines indicate the same bound at $\Lambda = 19$ and are included to illustrate the convergence. All bounds show a clear kink corresponding to the $O(N')$ model. An additional more dull kink is visible in the region $0.52 < \Delta_t < 0.58$. This kink gets less sharp and less precisely localized at larger $N$. For $N = 4$ an additional kink is visible around $\Delta_t = 1.1$. The bounds get strictly weaker for larger $N$.

## 3.1 Bounds on operator dimensions

**Singlets**

The bound on the dimension of the first singlet scalar $\Delta_S$ shows a clear kink corresponding to the $O(N')$ model under the identification $\phi^a \to t_{ij}$. In addition there is a second set of (dull) kinks in the region $0.52 < \Delta_t < 0.58$ whose exact location becomes less and less clear as $N$ increase. An additional kink is visible around $\Delta_t \approx 1.1$ for $N = 4$. These bounds are shown in figure 1. In the scalar singlet sector we do not find any new interesting feature.

Next, we explored bounds on $\Delta_{T'}$, the dimension of the first spin-2 singlet after the stress tensor. For small $N$ this bound shows a clear peak in the region $0.52 < \Delta_t < 0.58$. For larger $N$ the peak fades and the most discernible feature becomes a kink around $\Delta_t \approx 0.7$. However it seems that especially for larger $N$ the bounds are far from converged even at $\Lambda = 27$. These bounds are shown in figure 2.

It is a bit surprising that the bounds on the second spin-2 singlet are not very constraining. In fact, in most of CFTs based on a LGW description the next operator after the stress tensor has dimension $4 \lesssim \Delta_{T'} \lesssim 5$ [53, 54]. Similarly, in a gauge theory one expects to find an almost conserved spin-2 operator, coming from a combination the two stress tensors of the UV theory.[10] We believe these bounds are far from optimal: we will see an explicit example for the case $N = 4$ in the next section.

---

[10]In the limit of vanishing gauge coupling the theory contains two stress tensors, schematically $T_1^{\mu\nu} \sim \phi_i^\alpha \partial^\mu \partial^\nu \phi_i^\alpha$ and $T_2^{\mu\nu} \sim F^{\mu\rho} F_\rho^\nu$: in the IR one combination remains conserved while the orthogonal combination acquires an anomalous dimension.

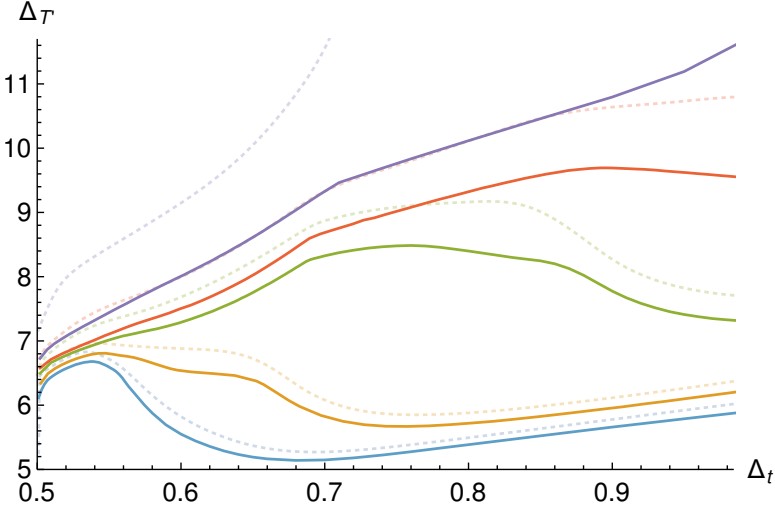

Figure 2: Bound on the dimension of the first spin-2 singlet after the stress tensor. The blue, orange, green, red and purple lines correspond respectively to $N = 4, 5, 10, 20, 100$. These bounds have been obtained at $\Lambda = 27$. The dotted lines indicate the same bound at $\Lambda = 19$ and are included to illustrate the convergence. For small $N$ a peak is visible. For larger $N$ the peak fades and the most discernible feature becomes a kink around $\Delta_t \approx 0.7$. The bounds get strictly weaker for larger $N$.

**Antisymmetric representation**

More interesting features are visible in the bound on the first spin-1 antisymmetric vector after the conserved current, shown in figure 3. This is the first instance where the bounds are neither strictly weaker nor stronger when increasing $N$. At large $\Delta_t$ we see the usual behavior found for singlet operators, i.e. the bounds get weaker for larger $N$. Near the unitarity bound the trend is instead reversed. The bounds start quite above the value expected in a GFT, which however doesn't contain a conserved current. For $N = 4, 5$ we observe a sudden drop of the bound (a reversed kink) followed by a smooth bound. For larger values the kink fades way, and a second bump appears for $N \sim 10$ close to the unitarity bound.
All the bounds diverge as $\Delta_t \to 1$ and for large values of $N$ an additional kink emerges.

The comparison of the bounds at $\Lambda = 19$ and $\Lambda = 27$ indicates a slow numerical convergence of the bounds for $\Delta_t \sim 1$, which get worse as $N$ increases.

**Box representation**

Next we examine the bound on the dimension of the first scalar Box operator, see figure 4. For small $N$ there are clear kinks in the region $0.54 \lesssim \Delta_t \lesssim 0.6$ . Additionally there is a family of very sharp kinks for all $N$ moving to the right towards $\Delta_t = 1$ as $N$ increases. In this case the location of the kinks is quite stable when passing from $\Lambda = 19$ to $\Lambda = 27$ and the bounds seem to be converged.

It would be tempting to identify the family of kinks at large $N$ with fixed points of gauge theories or $(A)RP^n$ models. Gauge theories discussed in section 1.3, however, are expected to contain operators with smaller dimension. On the other hand, $(A)RP^n$ are expected to have a large gap in this sector. In this case, one would expect $\Delta_t \sim 1 + O(1/N)$, while $\Delta_b \sim 4 + O(1/N)$. Unfortunately, the location of the kinks doesn't scale linearly with $1/N$, and it is unclear if they converge at all to $(\Delta_t, \Delta_b) = (1, 4)$ in the $\Lambda \to \infty, N \to \infty$ limit (see figure 18a in the appendix).

One possibility proposed in [55] is that bootstrap bounds for crossing equations based on

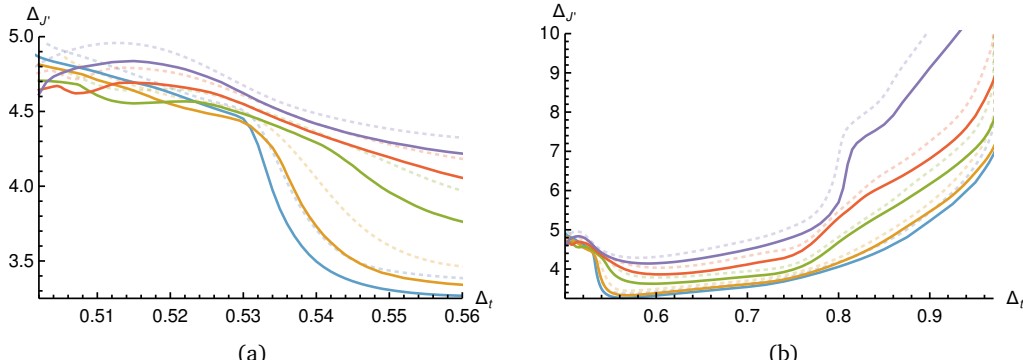

Figure 3: Both figures: Bound on the dimension of the first spin-1 antisymmetric vector after the conserved current. The blue, orange, green, red and purple lines correspond respectively to $N = 4, 5, 10, 20, 100$. The bounds have been obtained at $\Lambda = 27$. On the left: A zoom of the region $0.5 < \Delta_t < 0.58$. On the right: Overview of the same bound on $0.5 < \Delta_t < 1$. A second kink appears for $N = 10, 20, 100$ around $\Delta_t = 0.8$. The bounds diverge near $\Delta_t = 1$.

a symmetry $\mathcal{G}_N$ are in fact shaped by solutions with smaller symmetry $\mathcal{H}_M \subset \mathcal{G}_N$. This mechanism could explain the milder dependence on $N$: if for instance the expansion parameter of $\mathcal{H}_M$ is $1/M \sim 1/N^s$, with $s < 1$, then one would have a different scaling.

A different mechanism to produce kinks was proposed in [56]. In this case one could consider the difference between the 4pt function of a field $t_{ij} \sim \phi_i \phi_j + \dots$ made from two generalized free fields $\phi_i$ and the 4pt function of a generalized free field $\mathcal{T}_{ij}$. Since the former contains all the operators of the latter, it's possible to subtract the two 4pt functions and still have a decomposition in conformal blocks with positive coefficients. By subtracting the two, one can create large gaps and jumps in the bounds. This mechanism however would only explain kinks at $\Delta_t \geq 1$, as unitarity requires $\Delta_\phi \geq 1/2$.

**Hook representation**

A similar family of kinks can be seen in the bound on the dimension of the first spin-1 Hook vector as is shown in figure 5. However, the location of the kink in $\Delta_t$ does not precisely match the location of the kinks in the bound on the first scalar Box operator.

Again it would be tempting to identify these kinks with CFTs admitting a large-$N$ expansion but, as in the previous subsection, the dependence of the kink on $1/N$ doesn't seem to be linear or to converge to $(1, 4)$, at least at this value of $\Lambda$. In this case the situation is less clear, since the bounds seem farther from convergence in $\Lambda$, the features are less sharp, and they don't seem to strongly depend on $N$ for $N \geq 1000$.[11]

**Rank-2 tensor**

For $N > 2$ the OPE of two rank-2 symmetric tensors contains again rank-2 tensors. This offers the possibility to test the effect of a $\mathbb{Z}_2$ symmetry in the CFT. If $t_{ij}$ is odd under such a symmetry, then the 3pt function $\langle t t t \rangle$ must vanish. When inputting gaps on the rank-2 scalar sector above the external dimension $\Delta_t$, we then have the choice to allow the presence of an isolated contribution with $\Delta = \Delta_t$ or forbid it. This corresponds to the assumption that $t$ is respectively even or odd under a $\mathbb{Z}_2$ symmetry. We find strong evidence for a theory with a

---

[11]Neither the Hook nor the Box bound moves substantially when changing $N = 1000$ to $N = 10^{16}$ (this bound is not included in the figures).

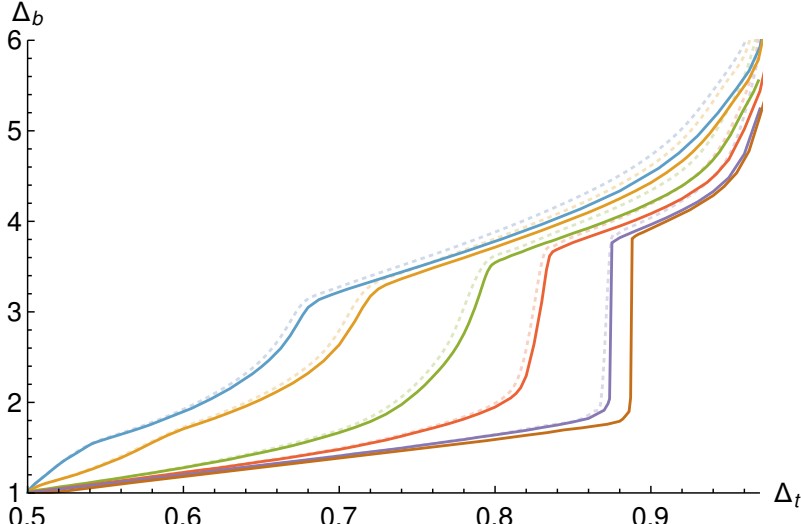

Figure 4: Bounds on the dimension of the first Box scalar. The blue, orange, green, red, purple and brown lines correspond respectively to $N = 4, 5, 10, 20, 100, 1000$. For $N = 4, 5$ there are kinks at $\Delta_t = 0.54$ and $\Delta_t = 0.60$ respectively. For larger $N$ this kink disappears. A family of sharp kinks is visible for all $N$.

$\mathbb{Z}_2$-even $t$ at large $N$. In figure 6 the bound on $\Delta_{t'}$ is shown both under the assumptions that $t \times t$ exchanges itself and without it. When we assume the exchange of $t$ itself in the $t \times t$ OPE, multiple sharp kinks appears for large $N$. The kink gets sharper as $N$ increases. Given the large values of $\Delta_{t'}$ at the kinks, we don't have plausible CFT candidates.

**Rank-4 tensor**

Finally, the bounds on the four-index-symmetric tensor are shown in figure 7. For small $N$ the only feature is the kink corresponding to the $O(N')$ model. For large $N$ a second kink emerges, for example at $N = 100$ a kink located around $\Delta_t \approx 0.82$.

The bounds continue smoothly for larger values of $\Delta_t$. If we assume accuracy of the value of $\Delta_t$ predicted for $O(N)$-vector models by large $N$ computations then these bounds force the presence in the spectrum of a relevant scalar for $N \gtrsim 10$.[12] The presence of this relevant operator makes the $O(N)$ models unstable with respect to (hyper)cubic perturbations. The same operator also drives the flow to the biconal fixed point with $O(m) \oplus O(N - m)$ global symmetry [57]. In [11] it was recently shown by numerical bootstrap applied to all correlators involving the first singlet, the first vector, and the first traceless symmetry scalar of $O(3)$ that $\Delta_{T^4} < 2.99056$. Thus, this operator is likely relevant for $O(N)$ models for all $N > 3$.

**External operator as the lowest dimensional operator of its kind**

There is one intuitive assumption that we have not used yet. We did not assume that $\Delta_{t'} \geq \Delta_{t_\text{ext}}$, i.e. that the external operator corresponds to the lowest dimensional traceless symmetric operator in the spectrum.[13,14] This assumption excludes for example a solution with both an operator $t'$ with $\Delta_{t'} = \Delta_{t_\text{ext}}$ and an operator $t$ with $\Delta_t < \Delta_{t'}$. However, the same

---

[12]Here we assume that the values predicted for $\Delta_t$ by the large-N expansion are reliable for these values of $N$ at the percent level.

[13]Thanks for Ning Su for bringing this to our attention.

[14]In this section we make a distinction between $\Delta_{t_\text{ext}}$ the dimension of the external operator $t_\text{ext}$ and the lowest dimensional or second lowest dimensional operators $t$ and $t'$ in a CFT solution.

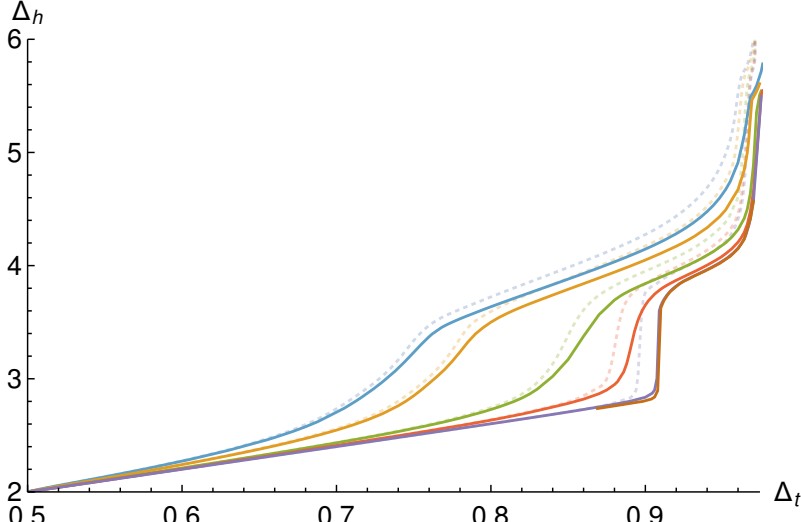

Figure 5: Bounds on the dimension of the first Hook vector. The blue, orange, green, red, purple and brown lines correspond respectively to $N = 4, 5, 10, 20, 100, 1000$. Again a family of sharp kinks is visible for all $N$. The locations of the kinks does not coincide with the family of kinks shown in the figure 4. The bounds have been obtained at $\Lambda = 27$.

solution also has to appear at $\Delta_{t_{ext}} = \Delta_t$. It is therefore not actually an additional assumption on the CFT. It merely keeps solutions from appearing twice at different values of $\Delta_{t_{ext}}$.

This can be generalized to the assumption that an external operator $\mathcal{O}_r$ is the $m$-th lowest dimensional operator in its representation $r$. Let's call this number an operator's *dimensional ordering number* $m$. Above we gave an example how in the setup studied in this paper we can impose that $t$ is the lowest dimensional traceless symmetric scalar in the CFT, i.e. $m = 1$. We can do this because the $t \times t$ OPE exchanges itself. In general this type of assumption can only be enforced if the representation of the external operator also appears as an exchanged internal operator. In that case we can instead also impose that the external operator corresponds to $m$-th lowest dimensional operator for any $m \in \mathbb{Z}_+$. However, this comes at the cost of having to scan over the dimensions of the $m - 1$ lower dimensional operators. If we do not impose any such condition at all we can only find the weakest bound among all these cases $m \in \mathbb{Z}_+$.

In figures 8a and 8b we show the effect of the $m = 1$ assumption on the bound on the dimension of the first Hook and Box scalars. For the region with $\Delta_t < 0.65$ this assumption does not lead to significant effects. However, in the region $\Delta_t > 0.65$ we find that the two families of kinks we found earlier move substantially. Importantly we now see that the family of kinks in the bound on lowest dimensional box operators asymptotes at large $N$ to the value expected in a large $N$ theory. Under this assumption the positions of the kinks seem to be well described by a $1/N$ expansion as can be seen in figures 18a and 18b. Moreover, a second family of (less pronounced) kinks, of which we previously could only see the $N = 4$ and $N = 5$ case becomes visible under this assumption. This family of kinks seems to asymptote towards $(1, 3)$.

For $N = 100$ this less pronounced kink also coincides with the large $N$ estimate of $\Delta_t$ in a theory with a global $O(N = 100)$ and a gauged $O(M = 4)$ symmetry [47]. Perturbatively a fixed point for such a theory is only expected to exist for $M = 1, 2, 3, 4$. Of these the $M = 4$ case is estimated to have the lowest value for $\Delta_t$ and should therefore be the first such theory to show up in our bounds (see equation 6).

The kinks in the bound on the first Hook scalar shown in figure 8b also show significant movement but still do not asymptote at large $N$ to a value predicted by a large $N$ limit. Some

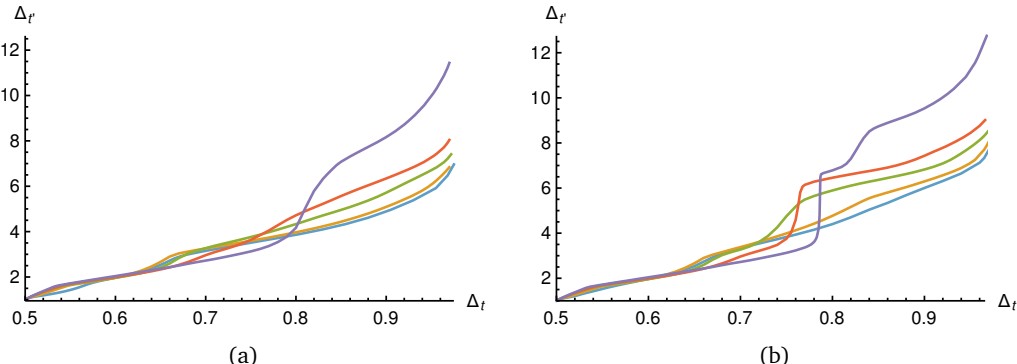

Figure 6: Bound on the dimension of the first traceless symmetric operator. The blue, orange, green, red and purple lines correspond respectively to $N = 4, 5, 10, 20, 100$. On the left: No additional assumptions. Various families of kinks are visible: One corresponding to the $O(N')$ model, one in the region $0.55 < \Delta_{t'} < 0.6$, one in the region $0.6 < \Delta_{t'} < 0.75$ (this one disappears at $N = 100$), and a last one in the region $0.75 < \Delta_{t'} < 1$. On the right: The same bound assuming that $t \times t$ exchanges $t$ itself. The last family of kinks becomes much sharper and more pronounced under this assumption especially for $N = 20, 100$. This is strong evidence that the kink corresponds to a theory with a $\mathbb{Z}_2$ even traceless symmetric operator. All bounds have been obtained at $\Lambda = 27$.

further assumption might be necessary to discover the "true" location of these kinks.

We also note that these bounds no longer diverge at $\Delta_{t_{\text{ext}}} = 1$. Instead they diverge around $\Delta_t = 2$. This is an important observation. It has been noted before that numerical bootstrap bounds often diverge when an external operator dimension approaches some integer value (see for example also [22]). The origin of some of these divergences can now be explained. For the Hook and Box bounds we see that the divergences can be removed by assuming that the external operator is the lowest dimensional operator of its type. Note that the divergence occurs exactly where we expect a new class of theories with $\Delta_{t'} = \Delta_{t_{\text{ext}}}$ to start to exist.[15] Given the existence of these theories it is thus not that surprising that in this region the $m = 2$ bounds dramatically weaken. This in turn implies that the bound where no assumption is made on the dimensional ordering number weakens at least as much.

Imposing the dimensional ordering number of the external operator could thus be an essential tool to exploring regions of large external operator dimensions.[16] Exploring larger values of $m$ increases the dimensionality of the search space and thus used to be prohibitively expensive. However using the new navigator method [58,59], this should now be feasible due to this method's superior scaling with the dimensionality of the search space.

One might also ask whether the position of the kinks we initially found could still be meaningful. Indeed a priori the kinks could still correspond to physically interesting CFTs with $\Delta_{t_{\text{ext}}} = \Delta_{t'}$. However, such a CFT would contain an operator $t$ with $\Delta_t < \Delta_{t'}$. In that case we would expect that the $t \times t$ OPE exchanges the same operators as the $t' \times t'$ operator (this holds even if $t$ is $\mathbb{Z}_2$-odd and $t'$ $\mathbb{Z}_2$-even or vice versa). That means that the bound found at $\Delta_{t_{\text{ext}}} = \Delta_t$ also applies to the spectrum exchanged in the $t' \times t'$ OPE. It is then easy to see from

---

[15]Think for example of free theories and generalized free theories where $\Delta'_t \geq 1$.

[16]Although this assumption does not eliminate all such divergences. Note that the bound on $\Delta_{J'}$ remains divergent at $\Delta_{t_{\text{ext}}} = 1$ even when we impose $m = 1$. Moreover in [22] similar divergences were observed even though there $m = 1$ was imposed there (in fact in that case a stronger condition was imposed since the $J \times \phi$ OPE **only** exchanges $\phi$ itself due to a ward Identity).

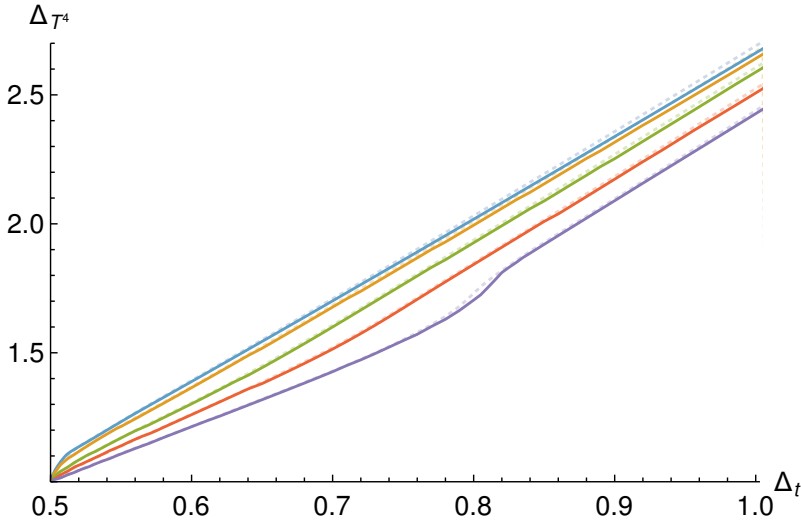

Figure 7: Bound on the dimension of the first four-index-symmetric scalar. The blue, orange, green, red and purple lines correspond respectively to $N = 4, 5, 10, 20, 100$. Apart from a kink at the location of the $O(N')$ model few features are visible. At $N = 100$ an additional kink becomes visible. These bounds have been obtained at $\Lambda = 27$. The dotted lines indicate the same bound at $\Lambda = 19$ and are included to illustrate the convergence. The bounds get strictly stronger for larger $N$.

our monotonically increasing bounds that this excludes the kinks we initially found and that they are thus unphysical.

## 4 Focusing on $O(4)$

Let us now focus on the case $N = 4$. This is the smallest $N$ we can discuss with the present formalism.[17] While it will be harder to compare against any large $N$ prediction, for this specific case there is a well defined candidate CFT to compare with. This is the $ARP^3$ lattice model studied in [45].

Our goal is to isolate an island in the OPE data corresponding to the $ARP^3$ model (or alternatively to exclude the existence of a plausible theory in the region predicted by lattice computations). Lattice computations find a fixed point with a traceless symmetric scalar with a dimension $\Delta_t = 0.54 \pm 0.02$ and exactly one relevant singlet with dimension $1.28 \pm 0.13$ [45].

We will first review the bounds presented in the previous section but zooming in on the region where the $ARP^3$ is expected to live. Next, we will also present a similar discussion about bounds on the OPE coefficients $\lambda_{ttT}$, $\lambda_{ttJ}$ and $\lambda_{ttt}$. Finally, we will choose a set of reasonable assumptions that allow to isolate the $ARP^3$ model.

### 4.1 Bounds on operator dimensions and OPE coefficients

Let us begin with the singlet sector. Unlike the Ising and $O(N)$ models for which precision islands have been previously obtained [5–7, 27, 61] the $ARP^3$ is not supposed to live close to the kink of the singlet bound. Instead it is predicted to lie well within the allowed region, see figure 9. As a consequence the theory is not easily isolated without making appropriate assumptions on the spectrum. However, we will see that bounds on other representations will

---

[17]The case of $O(3)$ is different because the OPE contains one representation less.



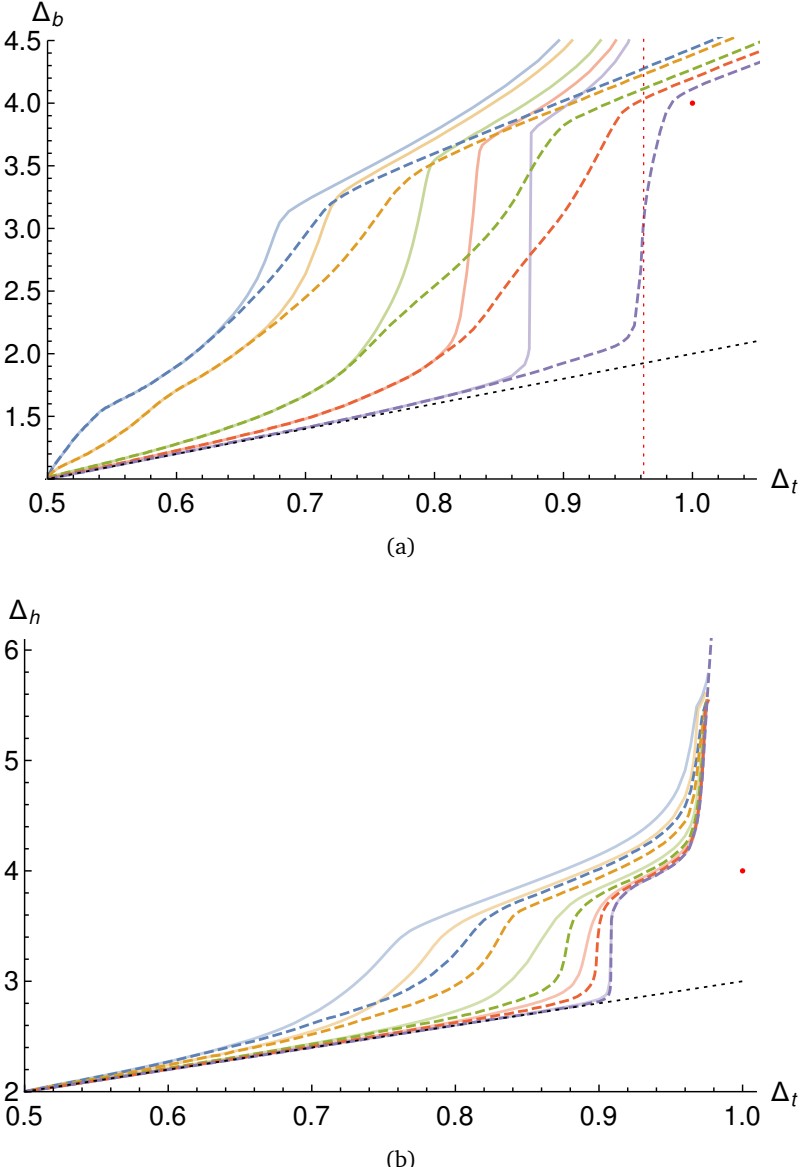

Figure 8: On the top: The dashed lines indicates the bound on the dimension of the first scalar Box operator under the assumption that $\Delta_t \geq \Delta_{t_{\text{ext}}}$. The bound without this assumption is also included for reference as a solid transparent line. The dashed lines show a family of sharp kinks assymptoting towards the point $(1, 4)$ (indicated by a red dot) where a large $N$ theory is expected to live. In addition a family of less pronounced kinks is also visible for all $N$, possibly assymptoting towards $(1,3)$. A red dotted line indicates the estimated $\Delta_t$ value in a theory with a global $O(N = 100)$ and a gauged $O(M = 4)$ symmetry. A black dotted line indicates the GFF family of solutions. On the bottom: The bound on the dimension of the lowest dimensional Hook scalar with (dashed) and without (solid) the assumption $\Delta_t \geq \Delta_{t_{\text{ext}}}$. The red dot indicates the position of a continuous gauge theory at large $N$ (see section 1.3). A black dotted line indicates the GFF family of solutions. The blue, orange, green, red and purple lines correspond respectively to $N = 4, 5, 10, 20, 100$. All bounds have been obtained at $\Lambda = 27$.

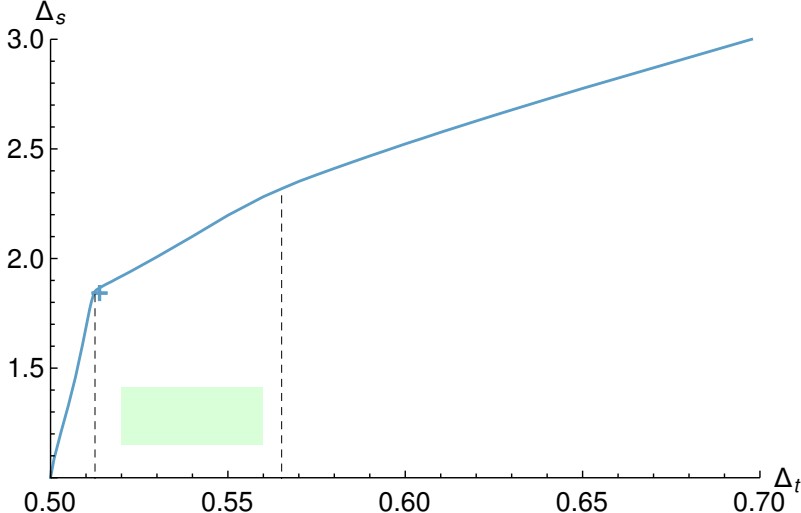

Figure 9: Bound on the dimension of the first singlet scalar. The black dashed lines indicate the positions of two kinks. The blue cross indicates the position of the O(9) model according to large $N$ estimates [60] (as seen from the traceless symmetric bootstrap under the identification $v^a \to v^{ij}$). The green region shows the prediction for the $ARP^3$ model from lattice computations. The bounds have been obtained at $\Lambda = 27$.

have features such as kinks and bumps which will drive our analysis.

Physical theories often stand out due to the presence of a large gap above known conserved operators [22, 62]. If we demand positivity on the stress tensor $T$ and maximize the gap $\Delta_{T'}$ until the next spin-2 neutral operator, we find a sharp peak as is shown in figure 10 (these bounds match those of the $O(N')$ vector bootstrap under the same assumption). The peak coincides with the lattice expectations for the location of the $ARP^3$ model. On the other hand a high value of $\Delta_{T'}$ is also expected close by due to the O(9) model at $\Delta_t \approx 0.519$, which is slightly before our region of interest.

Similarly the bound on $\Delta_{J'}$, the dimension of the first spin-1 antisymmetric vector after the conserved current, shows a clear feature within the region of interest. The kink in figure 11 hints at the existence of a theory with a high gap $\Delta_{J'}$ in the region $0.52 < \Delta_t < 0.535$.

Next we consider a bound on $\Delta_b$, the dimension of the first scalar Box operator. This bound shows two kinks[18] within the expected lattice region. This is shown in figure 12.

In the $ARP^3$ model the lowest dimensional traceless symmetric operator $t$ is expected to be odd under a $\mathbb{Z}_2$ symmetry, thus forbidding the exchange of $t$ itself in the $t \times t$ OPE. Thus, we should ask what the maximal allowed gap $\Delta_{t'}$ is. On the other hand, theories without a symmetry forbidding this exchange are expected to exchange $t$ itself as the first traceless symmetric operator. In that case we can assume the exchange of $t$ itself and bound the next traceless symmetric operator $t'$ by demanding positivity on $\Delta_t \cup [\Delta_{t'}^*, \infty)$. Both bounds are shown in figure 13. The first bound shows no special features in the region of interest. The second shows two kinks in the $ARP^3$ region. Also in the $ARP^3$ region the second bound is higher than the bound without this assumption. The two lines rejoin at a third kink outside the expected $ARP^3$ region (before separating again).

---

[18]One more pronounced, the other a mild change of slope.

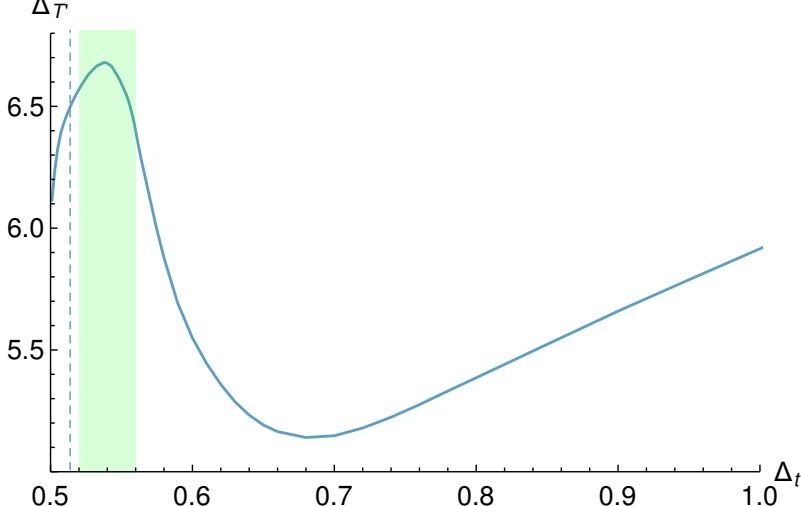

Figure 10: Bound on the dimension of the first spin-2 singlet after the stress tensor. The blue dashed line indicates the large $N$ estimate of $\Delta_\phi$ for the $O(9)$ model. The green region shows the prediction for the $ARP^3$ model from lattice computations. The bounds have been obtained at $\Lambda = 27$.

Finally for the sake of completion we show the bounds on the four-index symmetric scalar and the first Hook vector in figures 14a and 14b respectively. Neither of these bounds show any clear feature in the $ARP^3$ region.

We can also find lower and upper bounds on the OPE coefficients squared. An upper bound can be found for the OPE of any operator while a lower bounds can only be found if the operator is disconnected from other similar operators by a gap. We are mainly interested in the separable OPEs of the conserved operators $T$ and $J$. As usual the bounds on both of these OPE coefficients gets weaker for larger values of the external dimension. The $\lambda_{ttT}$ bound shows no clear features but the $\lambda_{ttJ}$ shows a kink around $\Delta_T = 0.535$. The value of the OPE found depends on the normalization of the conformal blocks (or equivalently the choice of normalization of the three and two point function) and thus it is often preferable to present the normalization invariant quantities of central charges divided by the value of the central charge in the free theory using the same normalizations. The resulting lower bounds on $C_T/C_{T_{\text{free}}}$ and $C_J/C_{J_{\text{free}}}$ are shown in figures 15a 15b.

In the next section we will try isolating island in the $(\Delta_t, \Delta_s)$ and $(\Delta_t, \Delta_b)$ planes using various assumptions. However, before we increase the dimensionality of the parameter space of our search it is smart to see how various assumptions influence the bisection bounds above.

For example, the Box operator shows one very strong kink in the regions allowed by the lattice bounds. However, by repeating that bound under the assumptions $\Delta_{T'} > 5.5$ and $\Delta'_J > 3$, we can see that simultaneously having both a high value near the top of the peak seen in figure 10 and high value near the plateau in figure 11 is incompatible with $\Delta_b$ taking a value close to this kink. This is shown in figure 19a. This is the first indication that perhaps the assumptions $\Delta_{T'} > 5.5$ and $\Delta_{J'} > 3$ are too strong. We will see more evidence for this later on. We can also consider how assumptions on $\Delta_{T'}$, $\Delta_h$ and $\Delta_b$ influence the maximal allowed gap $\Delta_{J'}$. An example of this is shown in figure 19b. This can be useful to already find the allowed $\Delta_t$ range under those assumptions in order to better locate any possible island in the larger spaces $(\Delta_t, \Delta_s)$ and $(\Delta_t, \Delta_b)$.

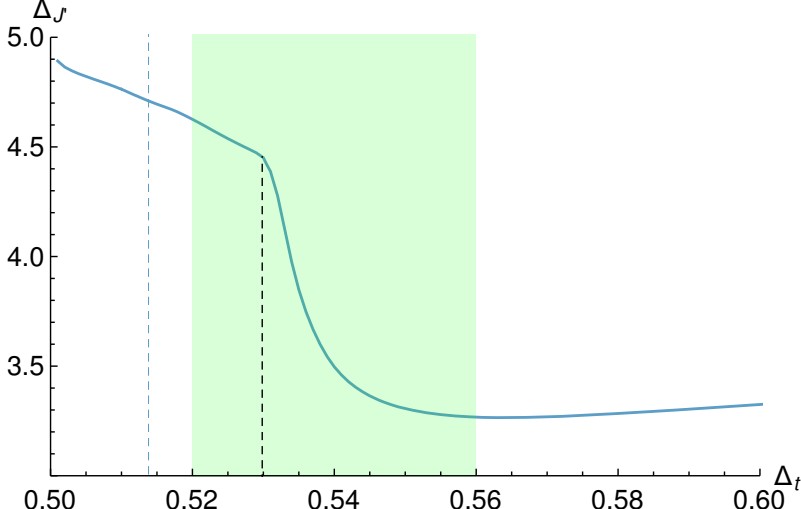

Figure 11: Bound on the dimension of the first spin-1 antisymmetric vector after the conserved current. The blue dashed line indicates the large $N$ estimate of $\Delta_\phi$ for the $O(9)$ model. The green region shows the prediction for the $ARP^3$ model from lattice computations. We see a clear kink within this region indicated by a black dashed line. In addition, various small kinks or wobbles appear in the region $0.51 < \Delta_t < 0.52$ though not in correspondence with large $N$ estimate of the location of the $O(9)$ model The bounds have been obtained at $\Lambda = 27$.

## 4.2 Isolating the $ARP^3$ model

In this section we report the results of our investigation. We present in this section only a few plots, and we refer to the appendix to support certain assumptions we make. Let us discuss them in order

1. Lattice simulations support the assumption that the model has a single relevant deformation and is not multi-critical. Unfortunately, assuming that $s$ is the only relevant scalar while $\Delta_{s'} > 3$ does not strongly narrow down the allowed region (see for instance figure 21a in the appendix). Thus we need to inject more assumptions.

2. In the previous section we observed a pronounced peak in the bound on the next operator after $T_{\mu\nu}$. In certain bootstrap studies, imposing a gap in this sector allows to create islands in the region of interest [22, 62]. In this case we tried several gaps: in the single correlator case considered so far, small gaps do not have any effect, while more aggressive gaps of $\Delta_{T'} \geq 5.5, 6.5$ create a small region, overlapping with the lattice prediction (see figure 23a). However, when considering mixed correlators those aggressive assumptions turn out to be completely disconnected from the lattice prediction or even ruled out (see figure 27). Thus we settled for the milder assumption $\Delta_{T'} \geq 4.5$.

3. A second strong feature was present in the bound on the first spin-1 antisymmetric operator after the $O(N)$ conserved current. Thus, we also add the assumptions that the first antisymmetric vector after the conserved current has a dimension larger than 3, i.e. assume that $\Delta_{J'}$ takes a value somewhere in the raised plateau in figure 11. This assumptions restricts the island further from the right and is compatible with the expected $ARP^3$ region (see an example in figure 23b).

4. In order to exclude the influence of the $O(9)$ model and the free theory with O(9) symmetry, it is useful to assume a small gap on the fist Hook vector dimension $\Delta_h$. Due to

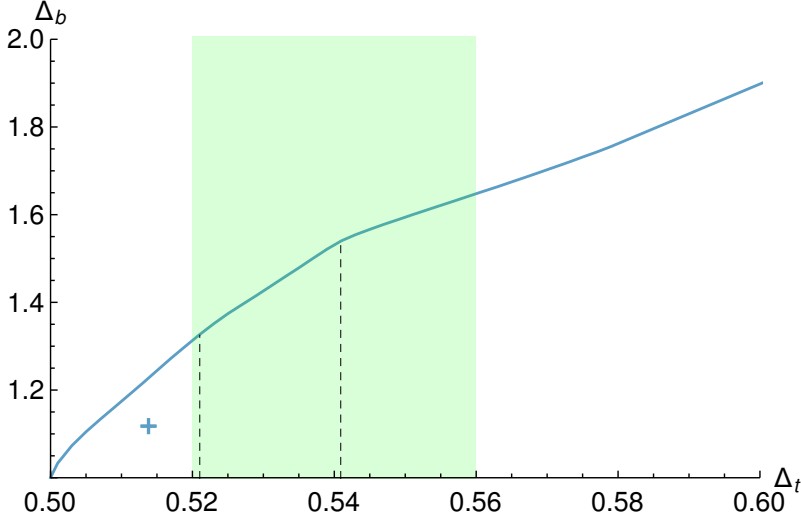

Figure 12: Bound on the dimension of the first scalar Box operator. The blue cross indicates the position of the O(9) model under the identification $v^a \to v^{ij}$, i.e. $(\Delta_v^{O(9)}, \Delta_t^{O(9)})$, according to large N estimates [60]. The green region shows the prediction for the $ARP^3$ model from lattice computations. There are two kinks in this region indicated here by black dashed lines. The bounds have been obtained at $\Lambda = 27$.

the identification $\phi^\alpha = t^{ij}$ and the resulting re-organization of operators these theories effectively have $\Delta_h = \Delta_J = 2$. Thus even a small gap above the unitarity bound can exclude these. Furthermore, no theory where the symmetry group really is O(4) is expected to have a conserved Hook vector. If we assume, for example, that $\Delta_h > 2+\delta$, with $\delta \sim 10^{-2}$ the peninsula detaches from theories with O(9) symmetry (see for instance figures 24, 25 and 26).

5. A final feature found in the previous section were two kinks in the bound on the first Box scalar. If we compute the allowed region in the $(\Delta_t, \Delta_b)$ plane assuming the existence of a single relevant box operator and the assumptions of point 2-4, we find an island in the neighborhood of the kink. Unfortunately the island disappears when pushing the numerics to $\Lambda = 35$ (see figure 22).If we relax the assumption of a single relevant box operator, the island survives at $\Lambda = 35$, see figure 16a in this section. The island is localized in the region $\Delta_b \geq 1.3$.

In conclusion, assumptions 1-5 allow to carve an island in the $(\Delta_t, \Delta_s)$ plane that overlaps with the lattice prediction and persists at $\Lambda = 35$. We show the result in figure 16b.

For the sake of completion we can investigate the existence of an island where the external $t$ is given by a $\mathbb{Z}_2$ even operator where $t$ itself is exchanged in the $t \times t$ OPE. Such a solution to the crossing equation is less likely to be a fake solution to crossing but it is also less likely to correspond to the $ARP^3$ CFT since $t$ is expected to be $\mathbb{Z}_2$-odd. In order to impose the exchange of $t$ we impose that the dimension of the first traceless symmetric operator after $t$ has a dimension $\Delta_{t'}$ greater than would be allowed without the exchange of $t$ itself, i.e. above the blue line shown in figure 13. This imposes the exchange of $t$ in $t \times t$, but this assumption also disallows theories exchanging $\Delta_t$ and an additional operator with $\Delta_{t'}$ both below the bound shown in figure 13. The resulting island is shown in 20. The persistence of the island means that we cannot exclude the island corresponding to a theory where $t$ is $\mathbb{Z}_2$ even.

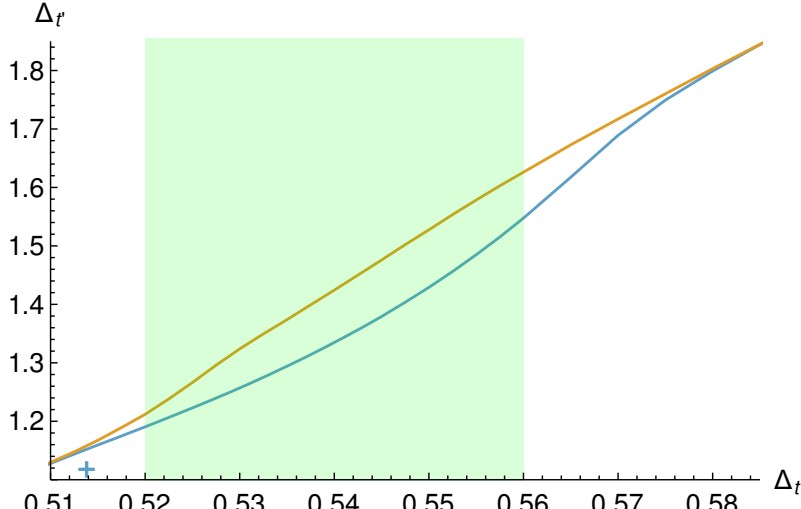

Figure 13: The blue line shows the bound on the dimension of the first traceless symmetric operator exchanged in the $t \times t$ OPE. The orange line shows the bound on the dimension of the first additional traceless symmetric operator $t'$ when assuming the exchange of $t$ itself. The blue cross indicates the large $N$ estimate of the position of the $O(9)$ model. The green region shows the prediction for the $ARP^3$ model from lattice computations. In the $ARP^3$ region allowing the exchange of $t$ itself lifts (weakens) the bound. The two lines join again at a third kink outside the expected $ARP^3$ region. The bounds have been obtained at $\Lambda = 27$.

## 4.3 Results mixed t-s bootstrap

In this section we report our investigation of the mixed correlator system of $t_{ij}$ and the leading scalar singlet $s$. In this setup we always have to scan over both $\Delta_t$ and $\Delta_s$. We assume the existence of a single operator with dimension $\Delta_s$, rather than a generic combination of operators with equal dimension. This is obtained by allowing a contribution with $\Delta_s$ in both $t \times t$ and $s \times s$ OPE and imposing a gap to the next scalar $\Delta_{s'} \geq 3$. In addition, we scan over the ratio of OPE coefficients $\{\lambda_{tts}, \lambda_{sss}\}$. The OPE scan was performed using the OPE scanning algorithm of Simpleboot [52]. Simpleboot efficiently takes advantage of the occurrences of both dual and primal jumps and the ability to hotstart SDPB from related points as well as the ability to exclude additional regions in the OPE space by solving a quadratic equation for the roots of the functional applied to the external vector contracted with generic ope coefficients, i.e. solving $\alpha(\{1, x\} \cdot V_{\text{ext}} \cdot \{1, x\}) > 0$ for $x$.[19]
The assumption that $s$ is the only relevant singlet has the net effect of restricting $\Delta_s > 1.052$.[20]

In the present setup we also have access to the mixed OPE, schematically

$$t \times s \sim t + t' + \dots \tag{20}$$

Previous bootstrap analysis of $O(N)$ models considered a scalar $\phi$ in the fundamental representation and studied mixed systems involving OPEs

$$\phi_i \times \phi_j \sim \mathbb{1} + s + s' + \dots, \qquad s \times s \sim \mathbb{1} + s + s' + \dots, \qquad \phi_i \times s \sim \phi_i + \phi_i' + \dots \tag{21}$$

In those cases, islands could be obtained by imposing the irrelevance of $s'$, and $\phi'$. In the present setup, instead, a similar assumption would exclude completely the $ARP^3$ region.

---

[19]We assumed $\lambda_{sss} \in \{-5000\lambda_{tts}, 5000\lambda_{tts}\}$. For example for the free theory $\lambda_{tts} = \lambda_{sss}$ in our normalization. Primal ratio's $\frac{\lambda_{tts}}{\lambda_{sss}}$ that we encountered were generally of order $O(1)$.

[20]This is expected since every critical (and not multi-critical system) was known to satisfy $\Delta_s > 1.044$ [13].

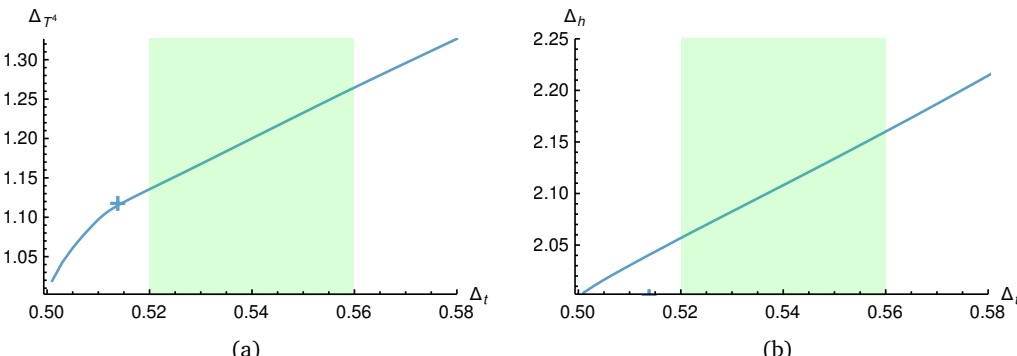

Figure 14: On the left: Bound on the dimension of the first four-index symmetric tensor. The blue cross indicates the large $N$ estimate of the position of the $O(9)$ model. Note that the estimate is excluded by these bounds, indicating an error due to higher order corrections and/or non-perturbative effects. On the right: The same plot but for the bound on the dimension of the first Hook vector. The green region shows the prediction for the $ARP^3$ model from lattice computations. Neither figure shows any features in this region. The bounds have been obtained at $\Lambda = 27$.

We can justify this behavior by considering the LGW model, although it does not predict a fixed point for $N = 4$. The Hamiltonian (3) contains two independent terms in the scalar potential. When imposing the equation of motion, one would become a descendant of $t$, while the orthogonal combination remains unconstrained. Thus, one naturally expects two relevant rank-2 scalars. In figure 17 we show the allowed region under these assumptions.[21] Unfortunately, they are not sufficient to create a closed island. In the $ARP^3$ region predicted by lattice simulation we find $\Delta_{t'} = 2 \pm 0.25$, while for larger $\Delta_s$ and $\Delta_t$ all values for $\Delta_{t'}$ are allowed.

In the same figure 17 we show a three dimensional extension of the island we found using the single correlator bootstrap (shown in figure 16a). In order to avoid a four dimensional scan we replace the assumption on $\Delta_{b'}$ with its resulting lower bound $\Delta_b > 1.3$. The use of the mixed correlator and OPE scan do not significantly shrink the $(\Delta_t, \Delta_s)$ space.[22]

## 5 Conclusions

In this work we initiated a bootstrap study of scalar operators transforming in $O(N)$ representations beyond the usual fundamental one. In particular we considered traceless symmetric rank-2 tensors $t_{ij}$. These operators are present in $O(N)$-vector model, with dimension $\Delta_t \sim 1 + O(1/N)$. In this work, however, we investigated an alternative situation, in which

---

[21] All bounds obtained using the mixed-correlator bootstrap are shown in orange to distinguish them from the single correlator bounds.

[22] When assuming a gap $\Delta_{T'}$ above the stress tensor in the mixed setup, we can enforce the ward identity $\lambda_{OOT} = \frac{\Delta_O}{\sqrt{C_T}}$. This is very effective, resulting in much stronger bounds than the equivalent single correlator bounds. Bounds corresponding to various assumptions on the gap $\Delta_{T'}$ are shown in figure 27. We find that the peak in $\Delta_{T'}$ that we found earlier (see figure 23a) was given by a fake solution since it disappeared by the addition of additional bootstrap equations (without making any additional assumptions). The new peak is no longer located in the expected $ARP^3$ region and is instead located at a much higher value of $\Delta_s$ and lies closer to the $O(9)$ model. Notably, the assumption $\Delta_{T'} > 5.5$ that we occasionally used in the previous sections is excluded for all $\Delta_s$ close to the lattice bounds. This suggests more caution is required when interpreting peaks and plateaus as evidence for a theory living high within that peak. Even so the "fake" peaks location is very suggestive and might still correspond to the location of the true $ARP^3$ model.

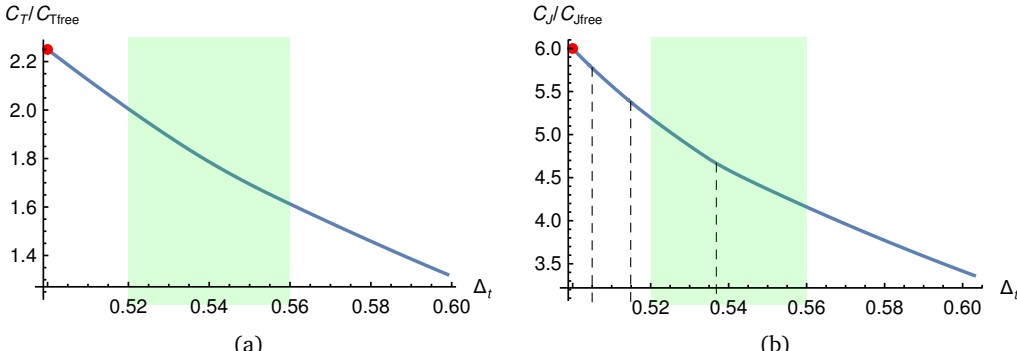

Figure 15: (a): Lower bound on $C_T$ in units of $C_{T_{\text{free}}}^{N=4}$. (b) Lower bound on $C_J$ in units of $C_{J_{\text{free}}}^{N=4}$. The Dashed lines indicate the locations of kinks in the upper bound on $\lambda_{ttJ}$. The red dots indicate the central charge values in the $N=9$ free vector boson theory. All bounds have been obtained at $\Lambda = 27$.

the operator $t$ plays the role of "elementary" (or smallest dimension) operator. This is the case for $(A)RP^{N-1}$ models, where it is the simplest gauge invariant operator, and in gauge theories with scalars in real representations.

A systematic study of the correlation function $\langle tttt \rangle$ for general $N$ revealed new and unexplained kinks. Most notably, two families of sharp kinks appear for all $N \geq 4$ in the bound on the first Box scalar and the first Hook vector. Additionally, we found various kinks in the bound on the dimension of the first traceless symmetric operator. Some of these kinks become much sharper if one assumes that the $t \times t$ OPE exchanges $t$ itself. We interpret this as evidence of CFTs where $t$ is even under any additional $\mathbb{Z}_2$ symmetry. This is the case for gauge theories and $RP^{N-1}$ models. Unfortunately none of the kinks agree with predictions obtained by large-$N$ expansions. Also, they do not seem to follow the expected pattern of anomalous dimension in a large-$N$ theory, i.e. $\gamma \sim O(1/N)$. We leave the investigation of these kinks (as well as some others described in the main text) to future research.

Next, we focused on the case $N = 4$, in the attempt to isolate a region corresponding to the phase transition observed in $ARP^3$ models by lattice simulations [63]. We found that simple assumptions, based on the number of relevant operators only, are unable to create an isolated region, not even after considering the mixed system of $t - s$ correlation functions. We found however a minimal set of assumptions able to carve out a closed region, overlapping with the lattice prediction.

By isolating a candidate island for the $ARP^3$ model this paper gives a partial answer to the discrepancy between the effective Landau-Wilson-Ginzburg description of $ARP^N$ model and their lattice simulations. The former predicts that no stable fixed points exist for $N > N_c \simeq 3.6$, while the lattice simulations show a clear second order phase transition. Possibly the perturbative estimate of $N_c$ is wrong despite it having a stable Padè-Borel approximation.

In order to settle completely this discrepancy it would very interesting to improve the analysis of [63] in order to extract additional information on other operators and compare them with the set of constraints on operator dimensions and OPE coefficients that we obtained for any CFT. In particular, we believe the scalar in the Box representation might play a fundamental role, see discussion in section 1.3. Moreover, by studying the pattern of symmetry breaking in the ordered phase, one can extract information about the signs of couplings in the LGW potential. Certain combination of signs could place the fixed point outside of the Borel summable region, thus explaining the tension.

Finally, there remains the possibility that the transition is actually first order with a large but

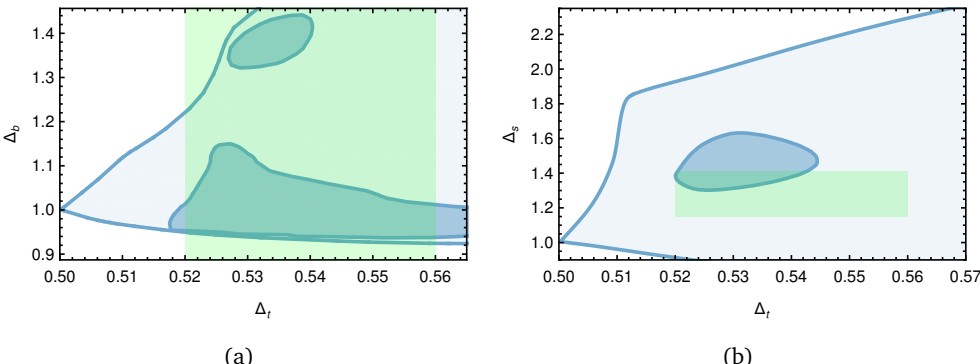

Figure 16: On the right: The light blue region shows the allowed region in the $(\Delta_t, \Delta_b)$ plane under the assumption $\Delta_{B'} > 3$. The dark region shows the allowed region under the assumptions $\Delta_{T'} > 4.5$, $\Delta_{J'} > 3$, $\Delta_h > 2.05$ and $\Delta_{B'} > 2.8$. The green region shows the prediction for the $ARP^3$ model from lattice computations. On the right: Corresponding allowed region in the $(\Delta_t, \Delta_s)$ plane (assuming $\Delta_b > 1.3$ instead of $\Delta_{b'} > 2.8$ to avoid scanning over a 3 dimensional parameter space). The bounds have been obtained at $\Lambda = 35$.

finite correlation function. We find it unlikely however that a complex CFT [64, 65] could produce the features observed in the bootstrap bounds presented in section 4.

It would also be interesting to repeat a similar analysis for higher values of $N$, looking for evidences of phase transition in $A(RP)^{N-1}$ models for generic $N$.

Alternatively, one could consider bootstrapping more correlation functions, along the lines of [10, 11].

# Acknowledgments

We thanks Ning Su for assistance in the installation and usage of the program Simpleboot. We also thank Slava Rychkov, Andrea Manenti, Andy Stergiou, Ettore Vicari and Claudio Bonati for useful comments and discussions. For most of the duration of the project MR and AV were supported by the Swiss National Science Foundation under grant no. PP00P2-163670. This project has received funding from the European Research Council (ERC) under the European Union's Horizon 2020 research and innovation programme (grant agreement no. 758903). M.R.'s research was also supported by Mitsubishi Heavy Industries (MHI-ENS Chair). All the numerical computations in this paper were run on the EPFL SCITAS cluster.

# A  2pt and 3pt functions

Instead of working with explicit indices, we contract all $SO(N)$ indices with suitable polarization vectors. As discussed in section 2, the OPE of two traceless symmetric representations contains generically mixed symmetry representations. In those cases, we will use different

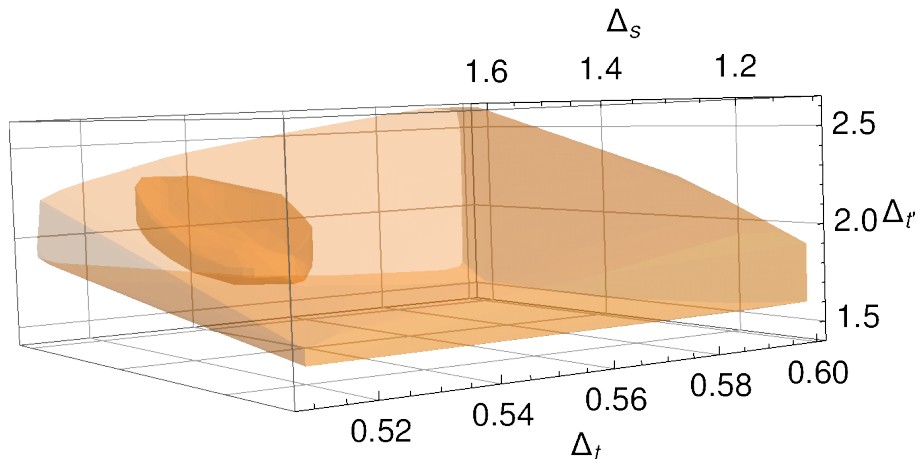

Figure 17: Allowed values for $\Delta_{t'}$ given $(\Delta_t, \Delta_s)$ in the expected $ARP^3$ region assuming the existence of exactly one relevant singlet and exactly one additional relevant $\mathbb{Z}_2$-odd operator besides $t$-itself. Darker: the same bounds under the additional assumptions: $\Delta_{T'} > 4.5$, $\Delta_{J'} > 3$, $\Delta_h > 2.05$ and $\Delta_b > 1.3$. The bounds have been obtained at $\Lambda = 19$.

polarization vectors, one for each antisymmetrized set of indices. Hence we will have:

$$\mathcal{O}_2(x,S) = \mathcal{O}_2^{ij}(x)S^iS^j\,, \qquad\qquad \mathcal{O}_4(x,S) = \mathcal{O}_4^{ijkl}S^iS^jS^jS^l\,, \qquad \text{(A.1)}$$

$$\mathcal{O}_{3,1}(x,S,U) = \mathcal{O}_{3,1}^{ijk,l}(x)S^iS^jS^kU^l\,, \qquad \mathcal{O}_{2,2}(x,S,U) = \mathcal{O}_{2,2}^{ij,kl}(x)S^iS^jU^jU^l\,, \qquad \text{(A.2)}$$

$$\mathcal{O}_{1,1}(x,S,U) = \mathcal{O}_{1,1}^{i,j}(x)S^iU^j\,. \qquad\qquad\qquad\qquad\qquad\qquad\qquad\qquad \text{(A.3)}$$

where we did not write the Lorentz tensor structure. At this point it is straightforward to compute the two and three point functions by imposing the correct symmetry (or antisymmetry) and traceless-ness properties. As usual, we can forget about the traceless-ness condition provided that we take all polarization vectors to be null: $S^2 = U^2 = 0$. The symmetrization of indices is also already taken care of by the contraction with the polarization vector. The only conditions left to impose are the antisymmetrization of indices corresponding to different lines of the Yang-tableau. This is easily expressed by the simple fact that if one replaces an $S$ vector with a $U$ vector, the result must vanish identically. More concretely the action of the differential operator $S \cdot \frac{\partial}{\partial U}$ must annihilate the expression. Moreover, if one contracts two antisymmetrized indices, the results vanishes too. In terms of polarization vectors, this means that the action of the differential operator $\frac{\partial}{\partial S} \cdot \frac{\partial}{\partial U}$ should gives zero as well.

Let us work out a simple example in details. The most general form of the two point function of a field $\mathcal{O}_{1,1}$ in the adjoint representation is

$$\langle \mathcal{O}_{1,1}(x_1,S_1,U_1)\mathcal{O}_{1,1}(x_2,S_2,U_2) \rangle = \mathcal{K}_2(x_{12})(a(S_1 \cdot U_1)(S_2 \cdot U_2) \qquad \text{(A.4)}$$

$$+ b(S_1 \cdot S_2)(U_1 \cdot U_2) + c(S_1 \cdot U_2)(S_2 \cdot U_1))\,. \qquad \text{(A.5)}$$

Imposing that both $S_i \cdot \frac{\partial}{\partial U_i}$ and $\frac{\partial}{\partial S_i} \cdot \frac{\partial}{\partial U_i}$ annihilate the above expression one can fix $a = 0$ and $b = -c$. We can take $b = 1$ for definitiveness.

Similarly one can get all other two point functions

$$\langle \mathcal{O}_0(x_1)\mathcal{O}_0(x_2)\rangle = \mathcal{K}_2^{\mathcal{O}}(x_{12}), \tag{A.6}$$

$$\langle \mathcal{O}_2(x_1,S_1)\mathcal{O}_2(x_2,S_2)\rangle = \mathcal{K}_2^{\mathcal{O}}(x_{12})(S_1\cdot S_2)^2, \tag{A.7}$$

$$\langle \mathcal{O}_4(x_1,S_1)\mathcal{O}_4(x_2,S_2)\rangle = \mathcal{K}_2^{\mathcal{O}}(x_{12})(S_1\cdot S_2)^4, \tag{A.8}$$

$$\langle \mathcal{O}_{1,1}(x_1,S_1,U_1)\mathcal{O}_{1,1}(x_2,S_2,U_2)\rangle = \mathcal{K}_2^{\mathcal{O}}(x_{12})\left((S_1\cdot S_2)(U_1\cdot U_2)-(S_1\cdot U_2)(U_1\cdot S_2)\right), \tag{A.9}$$

$$\langle \mathcal{O}_{3,1}(x_1,S_1,U_1)\mathcal{O}_{3,1}(x_2,S_2,U_2)\rangle = \mathcal{K}_2^{\mathcal{O}}(x_{12})\Big((S_1\cdot S_2)^3(U_1\cdot U_2)-$$
$$(S_1\cdot S_2)^2(S_1\cdot U_2)(U_1\cdot S_2)-\frac{2}{\mathcal{N}}(S_1\cdot S_2)^2(S_1\cdot U_1)(S_2\cdot U_2)\Big), \tag{A.10}$$

$$\langle \mathcal{O}_{2,2}(x_1,S_1,U_1)\mathcal{O}_{2,2}(x_2,S_2,U_2)\rangle = \mathcal{K}_2^{\mathcal{O}}(x_{12})\bigg((S_1\cdot S_2)^2(U_1\cdot U_2)^2 + (S_1\cdot U_2)^2(U_1\cdot S_2)^2$$

$$+\frac{2(S_1\cdot U_1)^2(S_2\cdot U_2)^2}{(N-1)(N-2)}-\frac{2(S_1\cdot S_2)(U_1\cdot U_2)(S_1\cdot U_1)(S_2\cdot U_2)}{(N-2)}$$

$$-2(S_1\cdot U_2)(U_1\cdot S_2)\left((S_1\cdot S_2)(U_1\cdot U_2)+\frac{1}{N-2}(S_1\cdot U_1)(S_2\cdot U_2)\right)\bigg). \tag{A.11}$$

Starting from the above definitions and using the Todorov operator acting on the $O(N)$ indices

$$\mathcal{D}_i(Z)=\left(\frac{N-2}{2}+Z\cdot\frac{\partial}{\partial Z}\right)\frac{\partial}{\partial Z^i}-\frac{1}{2}Z_i\frac{\partial^2}{\partial Z\cdot\partial Z}, \tag{A.12}$$

one can open the indices and obtain a tensor structure:

$$f_{i_1\ldots i_r}=\frac{1}{r!((N-2)/2)_r}\mathcal{D}_{i_1}(Z)\ldots\mathcal{D}_{i_r}(Z)f_{j_1\ldots j_r}Z^{j_1}\ldots Z^{j_r}. \tag{A.13}$$

For instance, one can obtain the three point function between two $t$ operators and an operator in the adjoint:

$$\left(\frac{2}{N-2}\right)^2(S_1\cdot S_2)(S_1\cdot\mathcal{D}_3(S))(S_2\cdot\mathcal{D}_3(U))[(S\cdot S_3)(U\cdot U_3)-(S\cdot U_3)(U\cdot S_3)]$$
$$=(S_1\cdot S_2)\left((S_1\cdot S_3)(S_2\cdot U_3)-(S_1\cdot U_3)(S_2\cdot S_3)\right). \tag{A.14}$$

Notice that here we had to add by hand a factor $(S_1\cdot S_2)$ to take care of the additional indices. For representations with four indices this is not needed. Similarly one can produce all the others. For example, starting from eq. A.11, we find the three point function between two $t$ operators and an operator in the Box representation:

$$\frac{4}{N^2(N-2)^2}(S_1\cdot\mathcal{D}_3(S))^2(S_2\cdot\mathcal{D}_3(U))^2[(S_1\cdot S_2)^2(U_1\cdot U_2)^2 + (S_1\cdot U_2)^2(U_1\cdot S_2)^2 + \ldots]$$
$$=((S_1\cdot U_3)(S_2\cdot S_3)-(S_1\cdot S_3)(S_2\cdot U_3))^2 + \frac{2}{(N-2)(N-1)}(S_1\cdot S_2)^2(S_3\cdot U_3)^2$$
$$-\frac{2}{N-2}(S_1\cdot S_2)(S_3\cdot U_3)((S_1\cdot U_3)(S_2\cdot S_3)+(S_1\cdot S_3)(S_2\cdot U_3)). \tag{A.15}$$

In a similar fashion one can also open the indices at point three and replace them with the polarizations of two other $t$ operators. This allows to create four point tensor structures.

# B   Four point tensor structures

Following the procedure outlined in Appendix A we are able to construct the tensor structures corresponding to each irrep exchange. Defining the basis:

$$
\begin{aligned}
B_1 &= (S_1 \cdot S_2)^2 (S_3 \cdot S_4)^2, & B_2 &= (S_1 \cdot S_2)(S_1 \cdot S_3)(S_2 \cdot S_4)(S_3 \cdot S_4), \\
B_3 &= (S_1 \cdot S_3)^2 (S_2 \cdot S_4)^2, & B_4 &= (S_1 \cdot S_2)(S_1 \cdot S_4)(S_2 \cdot S_3)(S_3 \cdot S_4), \\
B_5 &= (S_1 \cdot S_4)^2 (S_2 \cdot S_3)^2, & B_6 &= (S_1 \cdot S_3)(S_2 \cdot S_3)(S_2 \cdot S_4)(S_1 \cdot S_4),
\end{aligned}
\tag{B.1}
$$

Then the tensors structures become:

$$
\begin{aligned}
\hat{\mathbb{T}}_0 &= \frac{2}{(N+2)(N-1)} B_1, \\
\hat{\mathbb{T}}_2 &= \frac{2N}{(N+4)(N-2)} \left( B_4 + B_2 - \frac{2}{N} B_1 \right), \\
\hat{\mathbb{T}}_4 &= \frac{1}{6} (B_5 + B_3 + 4 B_6) - \frac{4}{3(N+4)} (B_4 + B_2) + \frac{4}{3(N+2)(N+4)} B_1, \\
\hat{\mathbb{T}}_{1,1} &= \frac{2}{N+2} (B_2 - B_4), \\
\hat{\mathbb{T}}_{3,1} &= \frac{1}{2} (B_5 - B_6) - \frac{2}{(N+2)} (B_4 - B_2), \\
\hat{\mathbb{T}}_{2,2} &= \frac{1}{3} (B_5 + B_3 - 2 B_6) - \frac{2}{3(N-2)} (B_4 + B_2) + \frac{2}{3(N-2)(N-1)} B_1.
\end{aligned}
\tag{B.2}
$$

In the above expression we have chosen the normalization such that the tensor structure are projectors and satisfy a completeness relation.[23] However, in order to keep the contribution of the identity operator with a simple normalization, we rescale:

$$
\hat{\mathbb{T}}_r = \frac{2}{(N+2)(N-1)} \mathbb{T}_r.
\tag{B.3}
$$

In this way $\mathbb{T}_0 = B_1$.

Similarly one can construct a tensor structure for the correlators $\langle tsts \rangle$, $\langle ttss \rangle$ and $\langle tsst \rangle$. In this case there is a single tensor structure. For $\langle ttss \rangle$ it is of the form

$$
\mathbb{T}_{ttss} = \frac{2N}{N-2} (S_1 \cdot S_2)^2,
\tag{B.4}
$$

and all the others can be obtained by crossing.

In the main text we considered a mixed system of correlators involving $\langle tttt \rangle$, $\langle ttss \rangle$ and $\langle ssss \rangle$. In order to connect the crossing equations resulting from the various correlators it is important to enforce the equality of OPE coefficients whenever possible. In the specific case, one would like to impose that the coefficient associated to the singlet exchange in the $t \times t$ OPE is the same, modulo the proper tensor structure, to the coefficients associated to the $t$ exchange in the $t \times s$ OPE. The formal way to ensure this would be to follow the procedure of [66]. Here we use a shortcut.

Let us begin defining the OPE coefficient

$$
\langle t(x_1, S_2) t(x_2, S_2) S(x_3) \rangle = \lambda_{ttS} (S_1 \cdot S_2)^2 \mathcal{K}_3(x_i, \Delta_i),
\tag{B.5}
$$

$$
\mathcal{K}_3(x_i, \Delta_i) = \frac{1}{|x_{12}|^{\Delta_1 + \Delta_2 - \Delta_3} |x_{13}|^{\Delta_1 - \Delta_2 + \Delta_3} |x_{23}|^{-\Delta_1 + \Delta_2 + \Delta_3}}.
\tag{B.6}
$$

---

[23]The sign of the projector gets fixed by imposing reflection positivity on the correlators in mirror symmetric configurations, see for example section III.E.1 in [3].

Next, we can compute this quantity in a solvable theory, for instance in a GFT, where

$$\langle \phi_i(x_1)\phi_j(x_2)\rangle = \frac{\delta_{ij}}{|x_{12}|^{2\Delta_\phi}}, \quad S = \frac{1}{\sqrt{2N}}\phi_i\phi_i, \quad t_{ij} = \frac{1}{\sqrt{2}}\left(\phi_i\phi_j - \frac{1}{N}\delta_{ij}\phi_k\phi_k\right). \quad \text{(B.7)}$$

Notice that while $S$ and $t_{12}$ are unit normalized, $t_{11}$ for instance is not. We obtain simply:

$$\lambda_{SSS} = \lambda_{ttS} = 2\sqrt{2/N}. \quad \text{(B.8)}$$

Finally we compute the correlation functions $\langle tttt\rangle$, $\langle ttss\rangle$ and $\langle tsts\rangle$ in GFT, single out the contribution of the conformal block associated to $t$ or $s$ and read-off the correct normalization of the tensor structures. In details:

$$K_{tttt}(x_i, \Delta_\phi)^{-1}\langle tttt\rangle\Big|_{\mathbb{T}_0} \supset 1 + \lambda_{ttS}^2(4r)^{2\Delta_\phi}(1 + O(r)),$$

$$K_{ttss}(x_i, \Delta_\phi)^{-1}\langle ttss\rangle\Big|_{\mathbb{T}_{ttss}} \supset 1 + \lambda_{ttS}\lambda_{SSS}(4r)^{2\Delta_\phi}(1 + O(r)),$$

$$K_{tsst}(x_i, \Delta_\phi)^{-1}\langle tsst\rangle\Big|_{\mathbb{T}_{tsst}} \supset \lambda_{tSt}^2(4r)^{2\Delta_\phi}(1 + O(r)).$$

One can check that the choice made in (B.2) and (B.4) are consistent.

## C $O(N)$ vs $O(N(N+1)/2-1)$ vector bootstrap.

When bootstrapping the system of equations for a $O(N)$ traceless symmetric operator the bounds on the dimension of the first singlet scalar are actually dominated by solutions related to $O(N')$ symmetry where $N' = N(N+1)/2 - 1$. The reason is that crossing equations for an $O(N')$ vector are related to those of an $O(N)$ traceless symmetric operator by an identification where the vector $\phi^a$ gets rewritten as $\phi^{ij}$ where $a \in \{0, ..., N'\}$ and $i, j \in \{0, ..., N\}$. The $\phi \times \phi$ OPE exchanges operators in the singlet (S), traceless symmetric (T) and antisymmetric (A) representations.[24] Any solution to the $O(N')$ vector bootstrap equations also solves the $O(N)$ traceless symmetric bootstrap equation (giving a solution with $\Delta_{T^2} = \Delta_{T^4} = \Delta_B = \Delta_T$ and $\Delta_{A^2} = \Delta_H = \Delta_A$).

Seen from the dual problem, one can show that there exist a positive linear map $T$ from any functional that is positive on the vectors $\{V_S, V_T, V_A\}$ to a positive functional on the vectors $\{V_S, V_{T^2}, V_{T^4}, V_{A^2}, V_H, V_B\}$. The resulting functional has the following (guaranteed) domain of positivity depending on the positivity properties of the original functional:

$$SO(N') : \alpha_v \to SO(N) : \beta_t, \qquad SO(N) : \beta_t \to SO(N') : \alpha_v,$$

$$\Delta_R^* \geq \begin{cases} \Delta_S^*, & R = S, \\ \Delta_T^*, & R \in \{T^2, T^4, B\}, \\ \Delta_A^*, & R \in \{A, H\}, \end{cases} \qquad \Delta_R^* \geq \begin{cases} \Delta_S^*, & R = S, \\ \max(\Delta_{T^2}^*, \Delta_{T^4}^*, \Delta_B^*), & R = T, \\ \max(\Delta_{A^2}^*, \Delta_H^*), & R = A. \end{cases} \quad \text{(C.1)}$$

Here $\Delta_R^*$ indicates the minimum of the domain of positivity, i.e. $\alpha(V_R) > 0 \,\forall\, \Delta \in [\Delta_R^*, \infty)$.

The proof below follows in the spirit of [55] were a similar relationship was proven between coinciding bounds in the bootstrap of $SU(N)$ fundamentals and the bootstrap of $O(2N)$ vectors.

---

[24]In this section T stands for the traceless symmetric representation appearing in the $\phi \times \phi$ OPE. We leave out the superscript in order to differentiate it from the traceless symmetric operators appearing in the $t \times t$ OPE. The same holds for the usage of A versus $A^2$.

**Theorem:** *Given a set of functionals $\alpha_a$ with $a \in \{1, ..., 3\}$ which are positive on respectively the three crossing equations of the O(N)-vector system, a set of positive functionals $\beta_i$ on the six bootstrap equations of the $O(N)$ traceless symmetric irrep can be found using positive linear map $T$ such that $\beta_j = \alpha_i T_{ij}$.*

**Proof:** The O(N)-vector equations can be written as

$$\sum_O \lambda_O^2 V_{S,\Delta,\ell} + \sum_O \lambda_O^2 V_{T,\Delta,\ell} + \sum_O \lambda_O^2 V_{A,\Delta,\ell} = 0_{1\times 6}, \tag{C.2}$$

or in matrix form as

$$M_{\langle vvvv\rangle, SO(N')} = \begin{pmatrix} 0 & F & -F \\ F & \left(1 - \frac{1}{N'}\right)F & F \\ H & -\left(\frac{1}{N'} + 1\right)H & -H \end{pmatrix} = 0, \tag{C.3}$$

where the rows correspond to the three different equations and the columns correspond to the vectors $V_S$, $V_T$ and $V_A$.

The problem of positive semi-definiteness of the bootstrap equation (after taking out the term corresponding to the unit operator) can be written as finding $\alpha_i$ such that

$$(\alpha_S \ \alpha_T \ \alpha_A) \equiv (\alpha_1 \ \alpha_2 \ \alpha_3) \cdot M_{\langle vvvv\rangle, SO(N')} \geq 0, \quad \forall \Delta_{R,\ell} > \Delta_{R,\ell}^*. \tag{C.4}$$

We will show the existence of $T_{ij}$ such that $\beta_j = \alpha_i T_{ij}$ and

$$(\alpha_S \ \alpha_{T^2} \ \alpha_{T^4} \ \alpha_{A^2} \ \alpha_H \ \alpha_B) \equiv (\beta_1 \ \beta_2 \ \beta_3 \ \beta_4 \ \beta_5 \ \beta_6) \cdot M_{\langle tttt\rangle, SO(N)} \geq 0, \quad \forall \Delta_{R',\ell} > \Delta_{R',\ell}^*. \tag{C.5}$$

Decomposing the irrep contributions $\{V_S, V_T, V_A\}$, according to the contributions to $\{V_S, V_{T^2}, V_{T^4}, V_A, V_H, V_{Box}\}$, one finds the following branching rules:[25]

| $\langle vvvv\rangle$ of $SO(N')$ | | $\langle tttt\rangle$ of $SO(N)$ | |
|:---:|:---:|:---:|:---:|
| $V_S$ | $\longleftrightarrow$ | $V_S,$ | (C.6) |
| $V_T$ | $\longleftrightarrow$ | $V_{T^2} + V_{T^4} + V_B,$ | (C.7) |
| $V_A$ | $\longleftrightarrow$ | $V_A + V_H.$ | (C.8) |

This motivates us to restrict our search to a map $T$ such that

$$(\beta_S \ \beta_{T^2} \ \beta_{T^4} \ \beta_{A^2} \ \beta_H \ \beta_B) = (\alpha_S \ x_1\alpha_T \ x_2\alpha_T \ x_4\alpha_A \ x_5\alpha_A \ x_3\alpha_T). \tag{C.9}$$

In other words we assume that the map $T$ relates the vectors $\beta_{R'}$ to $\alpha_R$ through $\beta_{R'} = \alpha_R \tilde{T}^R_{\ R'}$ with

$$\tilde{T} = \begin{pmatrix} 1 & 0 & 0 & 0 & 0 & 0 \\ 0 & x_1 & 0 & x_2 & 0 & x_3 \\ 0 & 0 & x_4 & 0 & x_5 & 0 \end{pmatrix}. \tag{C.10}$$

For this ansatz to hold the related linear transformation $T$ between $\alpha_i$'s and $\beta_i$'s has to be of the form

$$T = \tilde{T} \cdot M_{\langle tttt\rangle, O(N)}^{-1}. \tag{C.11}$$

---

[25] It is essential that $N' = \frac{N(N+1)-2}{2}$ for other $N'$ the branching of $T$ would also contain a singlet.

By imposing that the $F$ and $H$ equations do not mix we can fix the values $x_i$ and find a unique map $T$ (up to an overall constant). The values $x_i$ in this map are given by[26]

$$
\vec{x} = \frac{1}{N + N^2 - 4}\left( \frac{\left(\left(N+N^2\right)-2\right)^2}{N^2+N+2} \quad \frac{N(N+1)(N+2)(N+6)(N-1)^2}{12\left(N^2+N+2\right)} \right.
$$
$$
\left. \times \frac{N(N+1)(N+2)^2(N-3)(N-1)}{6\left(N^2+N+2\right)} \quad N(N-1) \quad \frac{1}{4}(N+1)(N+4)(N-2)(N-1) \right).
$$
(C.13)

The important thing to note is that these $x_i$ are positive for $n > 3$. Thus, any functional $\vec{\alpha}$ such that $(\alpha_S \ \alpha_T \ \alpha_A) \succcurlyeq 0$ guarantees that $(\beta_S \ \beta_{T^2} \ \beta_{T^4} \ \beta_{A^2} \ \beta_H \ \beta_B) \succcurlyeq 0$ since these are given by a positive coefficient times $\alpha_S$, $\alpha_T$ or $\alpha_A$. To be precise $\beta_S$ is guaranteed to be positive for $\Delta > \Delta_S$ while $\beta_{T^2}$, $\beta_{T^4}$ and $\beta_B$ are guaranteed to be positive for $\Delta > \Delta_T$ and $\beta_{A^2}$ and $\beta_H$ for $\Delta > \Delta_A$. (Positivity on this domain is guaranteed, but the functional can be positive on a bigger domain.)

Similarly an inverse map $T'$ can be found which provides a functional that is positive on $\{V_S, V_T, V_A\}$ from functionals positive on $\{V_S, V_{T^2}, V_{T^4}, V_A, V_H, V_{Box}\}$. In this case we look for a $T'$ such that

$$
(\alpha_S \ \alpha_T \ \alpha_A) = \begin{pmatrix} \beta_S & x_1\beta_{T^2} + x_2\beta_{T^4} + x_3\beta_B & x_4\beta_{A^2} + x_5\beta_H \end{pmatrix}.
$$
(C.14)

Again we find a unique solution for $T'$ and the parameters $x_i$

$$
x_i = \frac{4}{(n+n^2)-2}, \quad i = 1,2,3,4,5.
$$
(C.15)

Here we see that $\alpha_S \succcurlyeq 0$ is guaranteed when $\beta_S \succcurlyeq 0$, $\alpha_T \succcurlyeq 0$ is guaranteed to be positive on the domain where each of $\beta_{T^2}, \beta_{T^4}$ and $\beta_B$ are positive, i.e. $\Delta \geq \max(\Delta^*_{T^2}, \Delta^*_{T^4}, \Delta^*_B)$ and $\alpha_A \succcurlyeq 0$ is guaranteed to be positive if $\Delta \geq \max(\Delta^*_{A^2}, \Delta^*_H)$. The functional may be positive on a bigger domain. Thus, the (guaranteed) domains of positivity under the mappings $T$ and $T'$ are as described in equation C.1.

This means that the bootstrap equations of the $O(N)$ traceless symmetric scalar will gives the same bounds as the bootstrap of the vector equations of $O(N')$ as long as we assume positivity of the form $\Delta_{A^*} = \Delta_{H^*}$ and $\Delta_{T^{2*}} = \Delta_{T^{4*}} = \Delta_{B^*}$. However, stronger bounds can be found when we impose a different domain of positivity, i.e. different $\Delta_{O^*}$, for these operators.

---

[26] The explicit form of T in our normalization is given by

$$
T = \begin{pmatrix} 0 & \frac{\left(\left(N+N^2\right)-2\right)^2}{\left(N^2+N+2\right)\left(\left(N+N^2\right)-4\right)} & \frac{N(N-1)}{\left(N+N^2\right)-4} & \frac{N(N+1)(N+2)(N+6)(N-1)^2}{12\left(N^2+N+2\right)\left(\left(N+N^2\right)-4\right)} & 0 & 0 \\ 1 & \frac{\left(12-8N+2N^3+N^4\right)-7N^2}{\left(N^2+N+2\right)\left(\left(N+N^2\right)-4\right)} & -\frac{N(N-1)}{\left(N+N^2\right)-4} & \frac{N(N+1)(N+3)(N+6)(N-2)(N-1)}{12\left(N^2+N+2\right)\left(\left(N+N^2\right)-4\right)} & 0 & 0 \\ 0 & 0 & 0 & 0 & 1 & \frac{(2-N)-N^2}{\left(N+N^2\right)-4} \end{pmatrix}.
$$
(C.12)



# D   Additional Plots

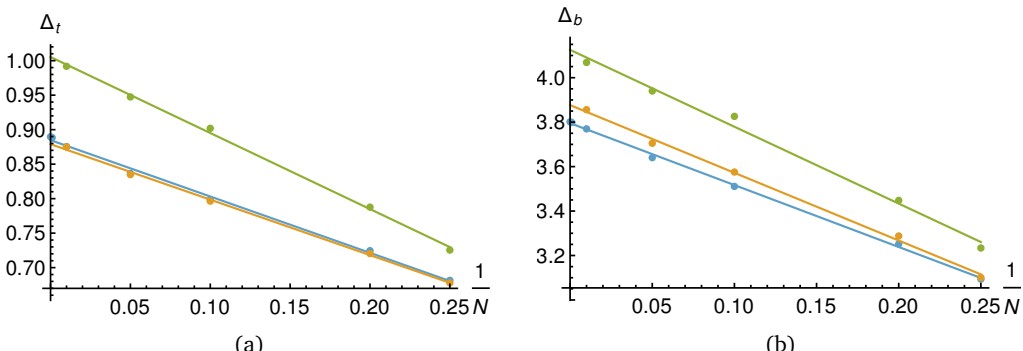

Figure 18: On the left: approximate $\Delta_t$ value of the kinks observed in the bound on the Box representation (figure 4) as a function of $1/N$. On the right: approximate $\Delta_b$ value of the kinks as a function of $1/N$. The yellow and blue dots corresponds to $\Lambda = 19, 27$ while the green dots are found under the additional assumption $\Delta_t \geq \Delta_{t_{\text{ext}}}$ see also figures 8a and 8b. The lines shows the best linear fit.

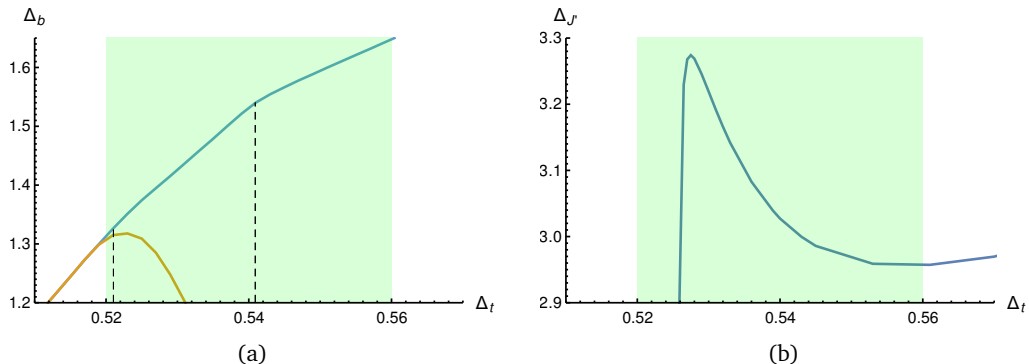

Figure 19: On the left: Bound on the dimension of the first scalar Box operator. The blue line shows the bound under no assumptions while the orange line is found under the assumptions $\Delta_{T'} > 5.5$ and $\Delta_{J'} > 3$. The orange line shows a maximum close to the position of the first kink of the blue line. The green region shows the prediction for the $ARP^3$ model from lattice computations. These bounds were obtained at $\Lambda = 19$ On the right: The bound on the first antisymmetric spin-1 operator after the conserved current assuming $\Delta_{T'} > 4.5$, $\Delta_h > 2.05$ and $\Delta_b > 1.37$. This bound has been obtained at $\Lambda = 35$.

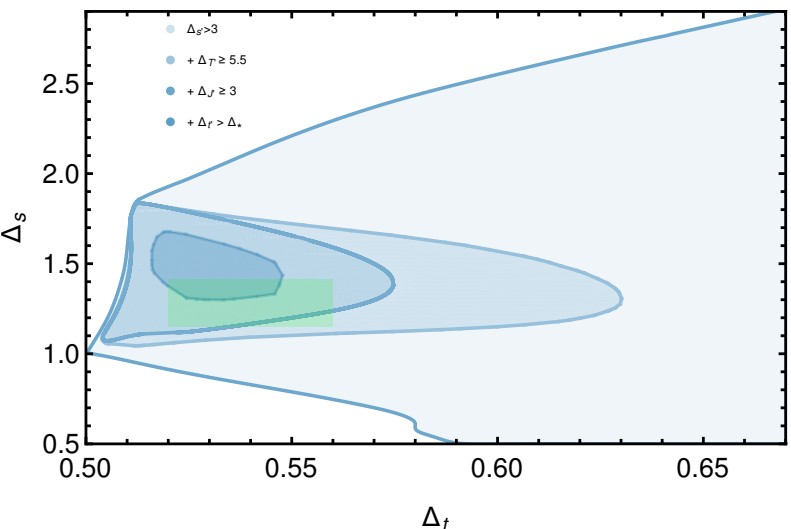

Figure 20: Allowed region in the $(\Delta_t, \Delta_s)$ plane assuming the existence of exactly one relevant singlet and successively more constraining assumptions as described in the legend. The assumptions $\Delta_{t'} > \Delta_*$ means that we allow the exchange of t itself but assume a gap $\Delta_{t'} > \Delta_*(\Delta_t)$ where $\Delta_*(\Delta_t)$ is the value of the upper bound found on $\Delta_{t'}$ without any additional assumptions(see figure 13). This assumption excludes all theories where $t$ itself is not exchanged (and hence should exclude the $ARP^3$ model. The green region shows the prediction for the $ARP^3$ model from lattice computations. The bounds have been obtained at $\Lambda = 19$.

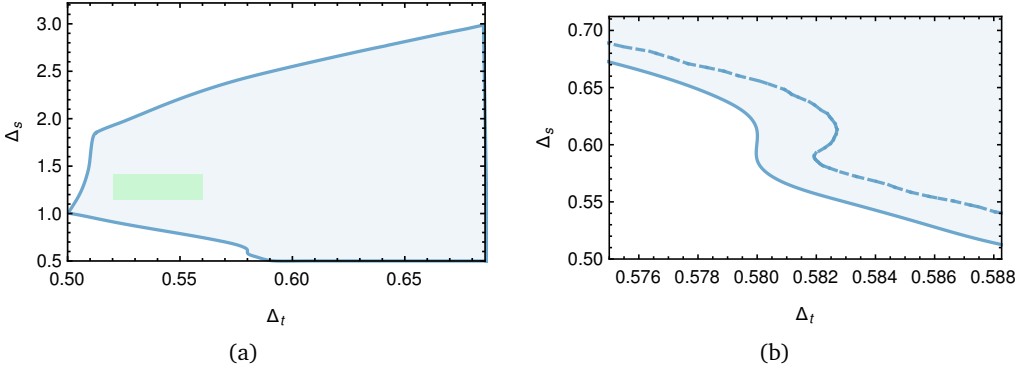

| (a) | (b) |

Figure 21: On the left: Allowed region in the $(\Delta_t, \Delta_s)$ plane assuming the existence of exactly one relevant singlet. The green region shows the prediction for the $ARP^3$ model from lattice computations. Three features stand out. The free theory can be found at the sharp corner of the peninsula near the unitarity bound. Another corner is controlled by the O(9) model. Lastly a small appendix can be seen around $\Delta_s = \Delta_t = 0.58$. The bounds have been obtained at $\Lambda = 19$. On the right: Zoom of the small appendix on the bottom. As $\Lambda$ is increased the appendix moves to the right. The bounds have been obtained at $\Lambda = 19$ (solid) and $\Lambda = 27$ (dashed).

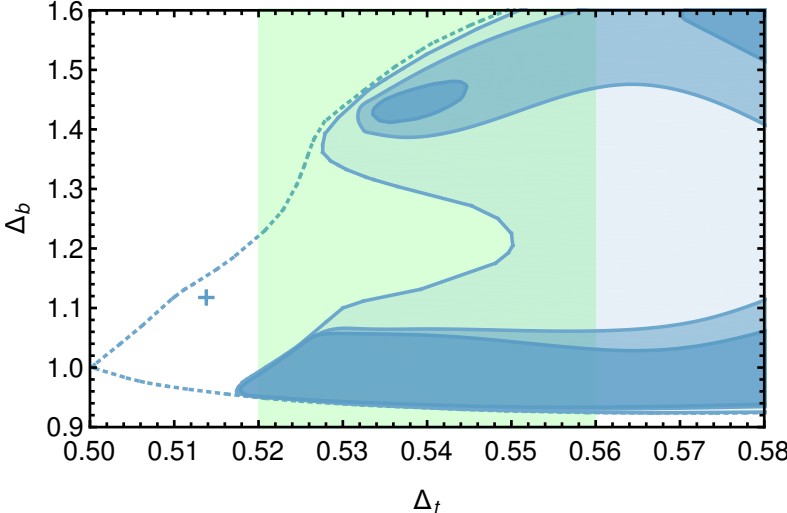

Figure 22: Allowed region in the $(\Delta_t, \Delta_b)$ plane assuming the existence of exactly one relevant Box scalar and $\Delta_{T'} > 4.5$, $\Delta_{J'} > 3$ and $\Delta_h > 2.05$. The green region shows the prediction for the $ARP^3$ model from lattice computations. The bounds have been obtained at $\Lambda = 19, 27, 31$ (light to dark). The isolated island disappears when we push to $\Lambda = 35$, indicating that at least one of these assumptions is too strong. The dashed line indicates the allowed region assuming only the existence of exactly one relevant Box scalar without additional assumptions.

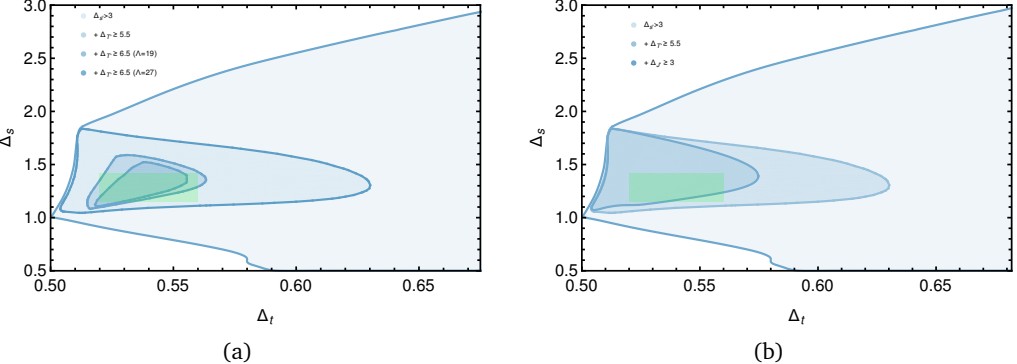

(a)  (b)

Figure 23: On the left: Allowed region in the $(\Delta_t, \Delta_s)$ plane assuming the existence of exactly one relevant singlet and $\Delta_{T'} > 5.5, 6.5$. The peak is clearly centered around the expected $ARP^3$ region. The bounds have been obtained at $\Lambda = 19, 27$ as indicated in the legend. On the right: Allowed region in the $(\Delta_t, \Delta_s)$ plane assuming the existence of exactly one relevant singlet and the gaps $\Delta_{T'} > 5.5$ and $\Delta_{J'} > 3$. Allowed regions under the lesser assumptions of one relevant singlet and the gap $\Delta_{T'} > 5.5$ are included for reference as described in the legend.

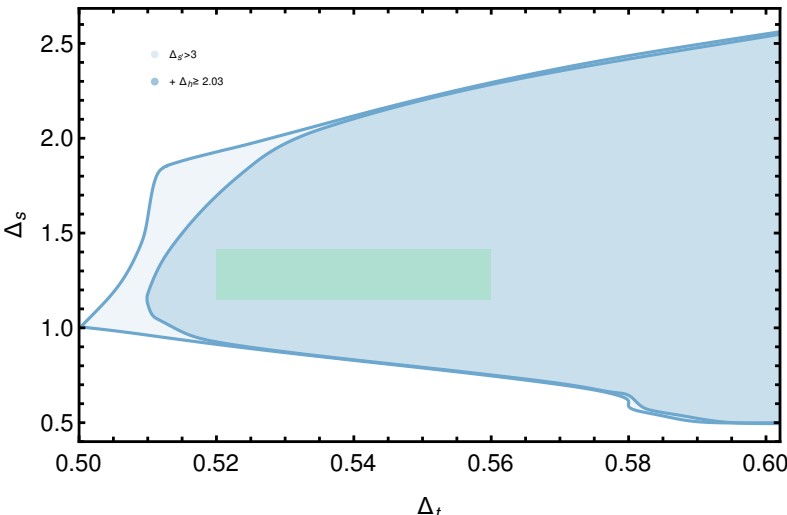

Figure 24: Allowed region in the $(\Delta_t, \Delta_s)$ plane assuming the existence of exactly one relevant singlet and $\Delta_h > 2.03$. The green region shows the prediction for the $ARP^3$ model from lattice computations. The bounds have been obtained at $\Lambda = 19$.

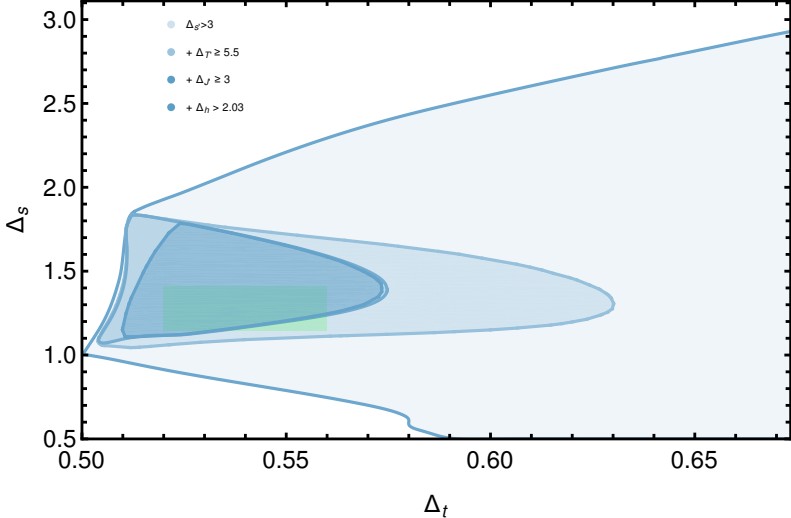

Figure 25: Allowed region in the $(\Delta_t, \Delta_s)$ plane assuming the existence of exactly one relevant singlet and $\Delta_{T'} > 5.5$, $\Delta_{J'} > 3$ and $\Delta_h > 2.03$. The green region shows the prediction for the $ARP^3$ model from lattice computations. The bounds have been obtained at $\Lambda = 19$.

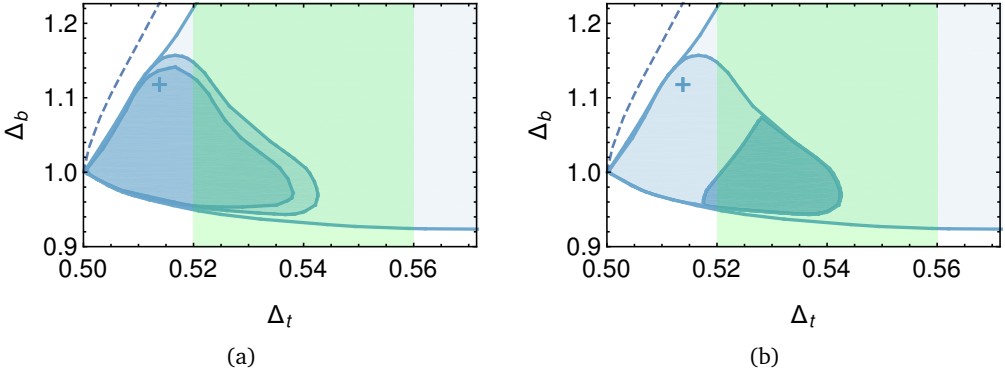

Figure 26: On the left: Allowed region in the $(\Delta_t, \Delta_b)$ plane assuming the existence of exactly one relevant Box scalar and $\Delta_{T'} > 5.5$, $\Delta_{J'} > 3$. The green region shows the prediction for the $ARP^3$ model from lattice computations. The blue cross indicates the large $N$ estimate of the position of the $O(9)$ model. The bounds have been obtained at $\Lambda = 19$ and $\Lambda = 27$. On the right: The same bound under the additional assumption that $\Delta_h > 2.05$. As expected this assumption seems to effectively exclude theories with $O(9)$ symmetry. The bound has been obtained at $\Lambda = 19$.

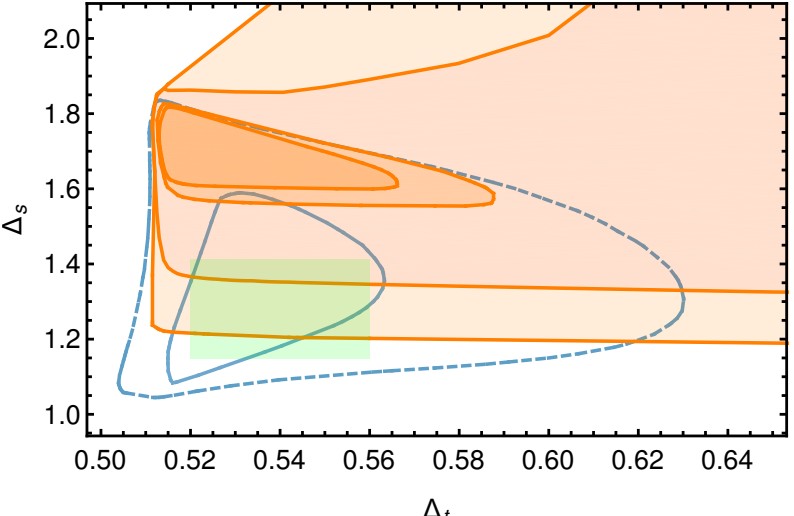

Figure 27: Allowed region in the $(\Delta_t, \Delta_s)$ plane assuming the existence of exactly one relevant singlet scalar and the gaps $\Delta_{T'} = 4.5, 5, 5.5, 5.6$ (light to dark). For reference the single correlator bounds under the assumptions $\Delta_{T'} > 5.5$ (dashed line) and $\Delta_{T'} > 6.5$ (solid line) are indicated in blue. In the mixed setup no primal points can be found for $\Delta_{T'} \geq 6$. The bounds have been obtained at $\Lambda = 19$.

# E   Parameters of the numerical implementation

The numerical conformal bootstrap problem was truncated according to the parameters in table 1. The semi-definite problem was solved using sdpb with the choice of parameters given in table 2.

Table 1: Values of the various parameters appearing in the numerical bootstrap problem.

|  | $\Lambda = 19$ | $\Lambda = 27$ |
| --- | --- | --- |
| Lset | $\{0, ..., 26\} \cup \{49, 50\}$ | $\{0, ..., 30\} \cup \{39, 40, 49, 50\}$ |
| order | 60 | 60 |
| $\kappa$ | 14 | 18 |

Table 2: Parameters used in sdpb for respectively feasibility problems and for OPE optimization.

| Parameter | feasibility | OPE |
| --- | --- | --- |
| maxIterations | 500 | 500 |
| maxRuntime | 86400 | 86400 |
| checkpointInterval | 3600 | 3600 |
| noFinalCheckpoint | True | False |
| findDualFeasible | True | False |
| findPrimalFeasible | True | False |
| detectDualFeasibleJump | True | False |
| precision | 700 | 700 |
| maxThreads | 28 | 28 |
| dualityGapThreshold | $10^{-20}$ | $10^{-20}$ |
| primalErrorThreshold | $10^{-60}$ | $10^{-60}$ |
| dualErrorThreshold | $10^{-60}$ | $10^{-60}$ |
| initialMatrixScalePrimal | $10^{20}$ | $10^{20}$ |
| initialMatrixScaleDual | $10^{20}$ | $10^{20}$ |
| feasibleCenteringParameter | 0.1 | 0.1 |
| infeasibleCenteringParameter | 0.3 | 0.3 |
| stepLengthReduction | 0.7 | 0.7 |
| choleskyStabilizeThreshold | $10^{-40}$ | $10^{-40}$ |
| maxComplementarity | $10^{200}$ | $10^{200}$ |

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
