# Peer review of "Bootstrapping traceless symmetric $O(N)$ scalars"

_SciPost Physics, doi:SciPost Phys. 14, 068 (2023)_

## Round 1 · Referee Report · Connor Behan (Referee 1) · 2022-9-19

Report

This work studies 3D critical points where the order parameter is a traceless symmetric tensor. This is the type of operator which led to a drastic shrinking of numerical bootstrap islands for the O(2) and O(3) models two years ago. For the present study, the authors shift their attention to the ARP3 model and other conjectured fixed points which are still associated with large error bars in the literature. A systematic search leads to an island which favours lattice predictions over those of the epsilon expansion.

This paper also communicates valuable lessons about how numerical bootstrap plots should be interpreted. Appendix C gives precise conditions which are required for the tensor bounds of one orthogonal group to differ from the vector bounds of another. There is also a nice discussion in section 3 showing that certain gaps may be imposed without loss of generality while still improving the bounds. The authors use this observation to find a new family of kinks, rule out an old family as unphysical and discuss future prospects involving the navigator method.

I have only good things to say about this paper's results but it seems to contain a number of typographical oversights which are listed below. I think at least a subset of them should be fixed before this proceeds to publication.

Requested changes

  1. Duplicated references are [10] = [62], [11] = [58] and [55] = [68].
  2. The paper frequently switches between "indexes" and "indices".
  3. It is often not consistent which letters are caligraphic. This includes O on page 31, N on page 32 and K on page 33. Similarly $Z_2$ on pages 16 and 27 should have its font changed.
  4. The last equation on page 31 uses the same dummy index too many times.
  5. The caption of Fig 18 neglects to mention that the green line comes from imposing that $t$ is the lightest symmetric tensor. Similarly, the caption of Fig 16 lists assumptions but it should mention that these were used for the dark region only.
  6. The right plot of Fig 19 lists four assumptions but $\Delta_{J^\prime} > 3$ makes less sense than the other three.
  7. Appendix A refers to "the previous section" instead of section 2 while the beginning of section 2.3 refers to section 4.3 instead of itself.
  8. On page 25, there is one reference to Fig 18b which should say 19b.
  9. At the top of page 18, there is a $t^\prime$ which should be a subscript of $\Delta$.
  10. Other changes to make are "a traceless symmetric tensors" -> "a traceless symmetric tensor" in the abstract, "the the O(2)" -> "the O(2)" on page 4, "the lattices size" -> "the lattice's size" on page 5, "will be denote by" -> "will be denoted by" on page 8, "due to this methods" -> "due to this method's" on page 19 and "simulation support" -> "simulations support" on page 25,
  11. Other missing plurals are "in many case" starting section 1.2 and "the above model have been" in section 1.3.

---

## Round 1 · Referee Report · Marco Serone (Referee 2) · 2022-10-6

The paper uses numerical conformal bootstrap techniques to put new constraints on 3d CFTs with $O(N)$ global symmetry. These are obtained by studying 4-point functions of scalars $t_{ij}$ in the rank 2 tensor representation of $O(N)$, assumed to be part of the CFT. After a general discussion on the bounds obtained for the lowest dimensional operators in the different OPE sectors for several values of $N$, the paper focuses on $N = 4$. This is motivated by the existence of a CFT associated to the so called $ARP^3$ model, confirmed by a lattice analysis but not confirmed by an $\epsilon$-expansion resummation. A mixed bootstrap analysis involving $t_{ij}$ and a singlet scalar is also considered.

The paper contains original and interesting results. The most notable ones are upper bounds on the scaling dimensions of operators in the representations exchanged in the $t \times t$ OPE for several values of $N$ and, for $N = 4$, the existence of a set of assumptions on the spectrum which gives rise to an allowed island in parameter space in the region of scaling dimensions found by the lattice results for the $ARP^3$ model. The authors provide also some technical comments on how to impose certain assumptions, which might be useful for future numerical analysis.

In section 1.3 there is an imprecision/inconsistency, which does not affect the main results of the work, but should nevertheless be fixed:

- In eq.(4) the index $a$ in $F_{\mu\nu}^a$ should run from 1 to $M(M-1)/2$ and not from 1 to $M$. The representation of $\phi_i^a$ under $SO(M)$ is never specified and there is an inconsistency in its choice. The covariant derivative in eq.(4), the formula in the text in the last row of page 6, and the form of $B_{ijk}$ in the next to last paragraph of section 1.3 in page 7 suggest that $\phi_i^a$ is in the adjoint representation. On the other hand, the text after eq.(4) indicates a fundamental representation. Similarly the large $N$ results of [48], eq.(5), apply for fields $\phi_i^a$ in the fundamental representation of $SO(M)$ (and fundamental of $O(N)$).

In addition to the typos already pointed out by the other referee:

1. In eq.(3), "$Tr$" $\rightarrow$ "Tr" in the second trace.

2. A final period is missing in footnote 3.

3. After eq.(7) "the the" $\rightarrow$ "the".

4. End of third paragraph in page 18: "assymptote" $\rightarrow$ "asymptote".

5. Beginning of section 4, page 19, " the the" $\rightarrow$ "the".

6. Page 27, paragraph after item 5, a final period is missing.

7. Caption figure 21: "$(\Delta_t, \Delta_t)$" $\rightarrow$ "$(\Delta_t, \Delta_s)$".

I am happy to recommend the paper for publication after the above imprecision has been fixed.

---

## Round 2 · Author Response

We made the final changes requested by Marco Serone in section 1.3

---

## Round 2 · List of Changes

Changed the example barion-like state and added footnote 5.

---

## Editorial Decision

published